

# Microbial diversity of the remote Trindade Island, Brazil: a systematic review

Glen Jasper Yupanqui García[1], Fernanda Badotti[2], Alice Ferreira-Silva[1], Joyce da Cruz Ferraz Dutra[1], Kelmer Martins-Cunha[3], Rosimeire Floripes Gomes[1], Diogo Costa-Rezende[4], Thairine Mendes-Pereira[1], Carmen Delgado Barrera[5], Elisandro Ricardo Drechsler-Santos[6] and Aristóteles Góes-Neto[1]

[1] Institute of Biological Sciences, Federal University of Minas Gerais, Belo Horizonte, Minas Gerais, Brazil
[2] Federal Center of Technological Education of Minas Gerais, Belo Horizonte, Minas Gerais, Brazil
[3] Federal University of Santa Catarina, Florianópolis, Santa Catarina, Brazil
[4] Department of Biology, Federal University of Ceará, Fortaleza, Ceará, Brazil
[5] University of San Francisco Xavier of Chuquisaca, Sucre, Bolivia, Sucre, Bolivia
[6] MIND.Funga/MICOLAB, Department of Botany, Universidade Federal de Santa Catarina, Florianópolis, Santa Catarina, Brazil

Corresponding author
Aristóteles Góes-Neto,
arigoesneto@gmail.com

## ABSTRACT

Trindade Island is a unique volcanic environment in the South Atlantic, characterized by acidic soils, rich organic matter and a high diversity of micro- and macroorganisms. Such diversity can represent a range of ecological niches and functions, potentially offering valuable ecosystem services. This systematic review aimed to synthesize the current knowledge of the island's microbial communities, focusing on their ecological roles and biotechnological potential. Following the PRISMA guidelines, a comprehensive search of the scientific literature was conducted to identify studies that performed DNA sequencing of samples collected on Trindade Island, Brazil. The selected studies used approaches, such as shotgun metagenomics and marker gene sequencing, including samples from microcosm experiments and culture-dependent samples. A total of eight studies were selected, but only six provided detailed taxonomic information, from which more than 850 genera of Bacteria, Archaea, and Fungi were catalogued. Soil communities were dominated by Actinobacteriota, Acidobacteriota, and Ascomycota (Fungi) while marine and coral environments showed high diversity of Pseudomonadota and Cyanobacteria. Microcosm experiments revealed adaptive responses to hydrocarbon contamination, mainly for Alcanivorax and Mortierella (Fungi). Compared to other ecosystems, such as the oligotrophic Galapagos Islands and the sea-restricted Cuatro Cienegas Basin, Cyanobacteria were shown to be more adaptive.

## INTRODUCTION

Trindade Island, the largest island in the Trindade-Martim Vaz Archipelago, is located in the South Atlantic Ocean at the same latitude as the municipality of Vitória, Espírito Santo, Brazil. The island has an area of approximately 10 km$^2$ and is 1,140 km from the coast

(*Camacho-Montealegre, Rodrigues & Tótola, 2019*; *Câmara et al., 2023*). The fertile soil of Trindade Island is shaped by a complex interplay of lithological, topographical, and biological activities. Lithological activities, which are driven by the island's volcanic origin, provide distinct minerals. Topographical activities favor the accumulation of organic matter in specific areas, while biological activities are driven by microorganisms that promote the biogeochemical cycles essential for the ecosystem maintenance (*Clemente et al., 2009*). The leaching of volcanic rocks provides trace elements to the marine and terrestrial ecosystems and influences the composition and diversity of microbial communities (*Costa-Rezende et al., 2023*; *Machado et al., 2017*). Studies conducted in other remote regions, such as Antarctica, indicated that variations in trace elements reflect changes in global atmospheric emissions, thereby reinforcing the unique characteristics of such environments (*Schwanck et al., 2019*).

Trindade Island is home to a diverse and unique biota, including endemic and endangered species (*Costa-Rezende et al., 2023*). Nevertheless, the island is confronted with a multitude of historical and contemporary environmental challenges that demand prompt intervention and efficacious strategies. The potential for pollution from a variety of sources also represents a significant threat to the island's ecosystems. These threats include rock erosion, volcanic processes, and sediment movement (*Cariou et al., 2017*; *Santos-Silva et al., 2018*), as well as the presence of plastic waste (*Andrades et al., 2018*; *de Souza Petersen et al., 2016*). For instance, soil contamination from oil pollution has been demonstrated to significantly alter the structure of the microbial community, reducing the number of fungal taxa and altering the composition of bacterial phyla (*Morais et al., 2016*). Furthermore, the introduction of goats from the 1700s to 2005 has also had a detrimental impact on soil quality and therefore, for soilborne microbial biodiversity (*Câmara et al., 2023*; *Costa-Rezende et al., 2023*; *Rodrigues et al., 2018*). The contamination by heavy metals such as zinc (Zn), lead (Pb), and copper (Cu) suggests an emerging scenario of pollution that alters microbial composition and compromises their essential ecological functions (*Santos-Silva et al., 2018*). The knowledge about the diversity and functioning of microorganisms in isolated environments, such as Trindade Island, is essential for predicting ecosystem resilience and developing conservation strategies.

Trindade Island has been studied for its microbial diversity (*e.g.*, *Câmara et al., 2023*; *Meirelles et al., 2015*) and biotechnological potential (*e.g.*, *da Silva et al., 2015*; *Rodrigues, Kalks & Tótola, 2015*; *Rodrigues et al., 2018*). Nonetheless, knowledge related to the diversity of Archaea, Bacteria, and Fungi on the island is limited (*Costa-Rezende et al., 2023*), especially on how the island's microbial communities respond to environmental stressors such as microplastic pollution, heavy metals, and other contaminants introduced into the marine and terrestrial environments. Moreover, there is an urgent need to further explore the structural diversity of the microbial communities of Trindade Island, as previous studies have shown that microplastic-biota interactions can alter microbial composition and functionality, affecting essential ecological processes (*Ivar Do Sul, Costa & Fillmann, 2014*; *Escalas et al., 2019*).

Archaea, Bacteria, and Fungi can represent unique species that perform specific functions and play key roles in biodiversity conservation (*Olofintila & Noel, 2023*). These microorganisms act as sensitive indicators of environmental contamination (*Korneykova et al., 2021*) and contribute to the protection of ecosystems (*Câmara et al., 2023*; *Rodrigues et al., 2018*) since they act in the degradation of pollutants and degradation of antibiotic through the production of specific enzymes (*Ghosh et al., 2020*). A systematic analysis of the current and most relevant studies conducted in Trindade Island allowed us to consolidate the available knowledge, and identify gaps and key areas for further studies. This review provides a comprehensive understanding of how microbial communities contribute to the ecosystem resilience, particularly in response to environmental stressors, such as pollution and climate change.

The aim of this study was to synthesize the ecological roles of Archaea, Bacteria, and Fungi previously identified in the terrestrial and aquatic environments of Trindade Island and their biotechnological potential.

## MATERIALS AND METHODS

### Data collection

This systematic review was based on the Preferred Reporting Items for Systematic Reviews and Meta-Analyses (PRISMA) guidelines (*Page et al., 2021*). Records were searched on 2 August 2024 in Scopus, Web of Science, PubMed, PubMed Central, Dimensions, and Google Scholar databases using the Publish or Perish v8 tool (*Harzing, 2007*). The keywords used to search for records were: "trindade island" AND (metagenomics OR shotgun OR (amplicons AND (16S OR 18S OR ITS OR "internal transcribed spacer"))). There were no restrictions on document type, language or publication date to avoid excluding relevant records.

The results of the database searches were exported in CSV format. The scripts *format_input.py* (*Jasper, 2023a*) and *remove_duplicates.py* (*Jasper, 2023b*) were used to select unique documents. These scripts read the CSV files and filter out DOI-less and duplicate documents from the set of records (*Dutra et al., 2023a*, *2023b*; *Gomes et al., 2023*; *Sierra et al., 2023*). The researchers García, G.J.Y. and Dutra, J.d.C.F. reviewed both the unique and DOI-less documents to select the scientific articles that met the selection criteria (Table 1). Disagreements in the selection of documents were resolved by the researcher Gomes, R.F.

After the initial review of titles and abstracts, only studies that addressed Trindade Island using shotgun metagenomics or amplicon sequencing were selected. These studies underwent a thorough evaluation, applying well-defined inclusion and exclusion criteria (Table 1), considering the collection area, sample type, methodologies used and the relevance of the results.

The following data were extracted from the selected documents: taxonomic abundance of microorganisms (Archaea, Bacteria, and Fungi) obtained by shotgun metagenomics and amplicon sequencing, type of substrate, samples or treatments, metagenomic approach, marker genes and sequencing platform used.

**Table 1 Eligibility criteria for the inclusion of articles in the systematic review.**

| Eligibility criteria | |
|---|---|
| Sampling location | Trindade Island, Brazil |
| Sample type | Any substrate |
| Sequencing type | Shotgun metagenomics, amplicons 16S rRNA, 18S rRNA or ITS |
| Focus taxa | Studies must focus on the domains Bacteria, Archaea, or the kingdom Fungi |
| Study | Original |
| **Exclusion criteria** | |
| Sampling location | Studies not conducted on Trindade Island, Brazil |
| Sample type | Studies that do not involve substrate samples or focus on human hosts |
| Sequencing type | Studies not using shotgun metagenomics, or amplicons for 16S rRNA, 18S rRNA, or ITS |
| Focus taxa | Studies that do not focus on Bacteria, Archaea, or Fungi, or that focus on other domains/kingdoms |
| Study | Review articles, meta-analyses, editorials, or other non-original studies |

## Extraction of taxonomic data

From the selected articles, taxonomic information on Archaea, Bacteria, and Fungi was extracted, both from the main text and Supplemental Materials. Considering that the selected articles used different databases and, when using the same databases, employed different versions, it was necessary to standardize the taxonomic nomenclature, identifying possible synonyms. For this, a local script developed in Python v3.10.8 accessed the Global Biodiversity Information Facility (GBIF) database (*GBIF Secretariat, 2023*) *via* its API. For taxa absent in the GBIF, manual searches were conducted using the NCBI Taxonomy database (*Schoch et al., 2020*).

## Grouping samples

The samples analyzed in the selected articles were collected from various sources, including soil, water, and coral tissue. These studies employed different sequencing approaches, including shotgun metagenomics and amplicon sequencing targeting the 16S rRNA and ITS markers. Additionally, two studies employing amplicon sequencing also conducted microcosm experiments prior to sequencing (*Morais et al., 2016*; *Rodrigues et al., 2018*).

To streamline the explanation of analyses in this review, samples were grouped based on similar characteristics. Samples subjected to sequencing without prior treatment were categorized with the suffix "**_Env**" (Environment), while those treated with microcosms were grouped under the suffix "**_MC**" (Microcosms). Nevertheless, in most of the statistical analyses, the sample itself was considered the unit of analysis.

The groups were defined as follows:

- **Soil_Env**: Independent-cultivation soil samples (PD5, PD6, PF7; see Table 2). Samples PD5 and PD6 were collected from the Pico do Desejado region, and sample PF7 was from the Fazendinha region, both located in the highest areas of Trindade Island.

- **Water_Env**: Independent-cultivation water samples (NOR_Island_W, PRI_Island_W, SAN_Island_W; see Table 2). The NOR_Island, PRI_Island, and SAN_Island samples were collected from Ponta Noroeste, Enseada do Príncipe (south), and Trindade Shelf (east), respectively.
- **Coral_Env**: Independent-cultivation coral tissue samples (FAR_Island_C, NOR_Island_C; see Table 2). The NOR_Island and FAR_Island samples were collected from Ponta Noroeste and Ponta dos Cinco Farilhões (southwest), respectively.
- **Soil_MC**: Soil sample treated with microcosms (Crude Oil; see Table 2). This soil sample was collected from the eastern region of the island.
- **Water_MC**: Twelve microcosm treatments using seawater samples (OIL, FLU, HEX, PHE, PHE+FLU, PHE+HEX, PHE+OIL, PHE+PYR, PYR; see Table 2). These water samples were collected from the southeastern part of the island, near Praia das Tartarugas.

*Rodrigues, Kalks & Tótola (2015)* and *da Silva et al. (2015)* identified a group of bacterial isolates through partial sequencing of the 16S rRNA gene using the MegaBACE 1,000 DNA Analysis System (SANGER) platform (Table 2). However, these studies did not provide abundance data, which is why this information was not included in the comparative analyses. It is worth noting that taxonomic classification at the genus and species levels was only achieved in articles using independent-cultivation sequencing techniques (Env). To facilitate spatial understanding of the samples, a static map and an interactive map using the OSM (OpenStreetMap) platform and the JavaScript Leaflet library were created.

## Analyses of the communities of Archaea, Bacteria, and Fungi

A series of sample analyses were performed using the programming language R v4.2.1. Relative abundances of all taxonomic categories were analyzed through bar charts generated with the ggplot2 v3.4.3 package (*Wickham, 2011*). Core (shared) and satellite (unique) taxa were examined using UpSet diagrams, generated with the UpSetR v1.4.0 package (*Conway, Lex & Gehlenborg, 2017*).

Alpha and beta diversity analyses were conducted based on the abundance at the order level, as it was the category where data was available for all samples. Shannon indices were calculated using the vegan v2.6.4 package (*Dixon, 2003*) and plotted with ggplot2 v3.4.3, based on relative abundances. Principal coordinate analyses (PCoA) were generated using the vegan v2.6.4, ggrepel v0.9.3 (*Slowikowski et al., 2018*) and ggplot2 v3.4.3 packages, based on Hellinger transformations.

Strong association networks between groups and taxa were calculated using Hellinger transformations with the indicspecies v1.7.14 package (*De Caceres, Jansen & De Caceres, 2016*). The multipatt function with the point biserial correlation coefficient (r.g) was used to calculate association strength, considered statistically significant at $p < 0.05$. Networks were constructed using Cytoscape v3.10.1 software (*Shannon et al., 2003*) with the edge-weighted spring-embedded layout. In the networks, circles represent taxa, the size of the circles indicates taxa abundance, diamonds represent sample groups, and the length of

**Table 2 Studies included in the systematic review separated by taxa studied, site investigated, molecular approach, and marker used for each sample.** Articles were published from 2014 to 2023.

| Article | Taxa studied | Substrate | Samples/Treatments | Study method | Approach | Marker gene | Primers | Analysis tool | Sequencing platform |
|---|---|---|---|---|---|---|---|---|---|
| *Meirelles et al. (2015)* | Archaea Bacteria Eukaryota (Fungi, Metazoa, Protozoa, Viridiplantae) Viruses | Water and Coral tissue | NOR_Island_W, PRI_Island_W, SAN_Island_W, FAR_Island_C, NOR_Island_C | Culture-independent (Env) | WGS metagenomics | – | – | MG-RAST | 454 GS FLX Titanium |
| *Câmara et al. (2023)* | Archaea Bacteria Eukaryota (Fungi, Metazoa, Protozoa, Chromista, Viridiplantae) | Soil | PD5, PD6, PF7 | Culture-independent (Env) | Amplicon | 16S rRNA (V3-V4), ITS | 341F-805R and ITS3-ITS4 | QIIME2 | Illumina MiSeq |
| *Rodrigues et al. (2018)* | Archaea Bacteria | Water | Water sample/OIL, FLU, HEX, PHE, PHE+FLU, PHE+HEX, PHE+OIL, PHE+PYR, PYR | Microcosms (MC) | Amplicon | 16S rRNA (V4) | 515F-806R | Brazilian Microbiome Project | Ion Torrent PGM |
| *Morais et al. (2016)* | Archaea Bacteria Fungi | Soil | Crude Oil | Microcosms (MC) | Amplicon | 16S rRNA (V4), ITS | 515F-806R and ITS1F-ITS2 | Brazilian Microbiome Project | Illumina MiSeq |
| *Rodrigues, Kalks & Tótola (2015)* | Bacteria | Water | Water sample/TRH1, TRH2, TRH3, TRH4, TRN1, TRN2, TRN3, TRN4, TRN5, TRN6, TRN7, TRN8, TRN9, TRN10, TRN11 | Culture-dependent (DCul) | Amplicon | 16S rRNA (V6-V8) | 005F-531R | BLASTn/ GenBank | MegaBACE 1000 DNA Analysis System |
| *da Silva et al. (2015)* | Bacteria | Soil | P01, P02, P03, P04, P05, P06, P07, P08, P09, P10, P11, P12/TR7, TR8, TR10, TR12, TR13, TR14, TR17, TR19, TR22, TR27, TR27II, TR35II, TR47II, TR59II | Culture-dependent (DCul) | Amplicon | 16S rRNA (V3-V4) | 5F-531R | BLASTn/ GenBank | MegaBACE 1000 DNA Analysis System |
| *Pylro et al. (2014)* | Archaea Bacteria | Soil | P04, P09, P11 | Culture-independent (Env) | Amplicon | 16S rRNA (V4) | 515F-806R | QIIME/USEARCH | Illumina MiSeq/Ion Torrent PGM |
| *Câmara et al. (2022)*[a] | Archaea Bacteria Eukaryota (Fungi, Metazoa, Protozoa, Chromista, Viridiplantae) | Soil | PD5, PD6, PF7 | Culture-independent (Env) | Amplicon | 16S rRNA (V3-V4), ITS | 341F-805R and ITS3-ITS4 | QIIME2 | Illumina MiSeq |

**Note:**
[a] Article excluded from the discussion because it is equivalent to the research by *Câmara et al. (2023)*.

edges represents the association strength, with shorter edges indicating stronger sample-taxon associations.

Non-parametric multivariate analyses were conducted using the PERMANOVA test and the Bray–Curtis dissimilarity index with 9,999 permutations in PAST v4.11 software (*Hammer & Harper, 2001*). The aim was to compare community structure among sample groups based on the Hellinger transformation. Additionally, pairwise PERMANOVA models were used to evaluate statistically significant differences between pairs, with a significance level of $p < 0.05$.

## RESULTS

### Study selection

Search protocols across databases identified a total of 274 studies (Scopus ($n = 24$), Web of Science ($n = 1$), PubMed ($n = 2$), PubMed Central ($n = 13$), Dimensions ($n = 98$), and Google Scholar ($n = 136$)) (Data S1, Data S2). Among these, 67 duplicate records were automatically removed prior to the initial screening, resulting in 207 articles being evaluated. During screening, six documents were excluded for lacking a DOI and having ambiguous or incomplete titles, leaving 201 unique articles for the eligibility analysis.

After a manual review of the methodologies and application of inclusion and exclusion criteria, 193 articles were discarded by researchers García, G.J.Y. and Dutra, J.d.C.F., who reached agreement on the selected documents, with only two cases of disagreement. These were resolved by researcher Gomes, R.F., who accepted one document and rejected the other for not meeting the selection criteria (Table 1). Ultimately, eight documents met the established criteria (Fig. 1, Table 2).

It is worth noting that the studies by *Câmara et al. (2022, 2023)* and used the same samples and presented equivalent results. As such, the *Câmara et al. (2022)* study was excluded from the discussion for being redundant, but it remains part of the table of the eight selected articles.

### Profile of selected articles

*Meirelles et al. (2015)* collected seawater and coral tissue samples, analyzing them through shotgun metagenomics. *Câmara et al. (2023)* collected soil samples and performed analyses using amplicon sequencing of 16S rRNA and ITS markers. *Rodrigues et al. (2018)* collected seawater samples and established nine microcosms (closed systems created from these samples) with the addition of various hydrocarbons. These systems were monitored for temperature, time, and hydrocarbon degradation and analyzed through 16S rRNA amplicon sequencing. *Morais et al. (2016)* collected soil samples, set up microcosms with the addition of aged crude oil, and subsequently performed analyses using 16S rRNA and ITS amplicon sequencing.

The majority of the selected studies focused on the bioremediation of hydrocarbon contamination (*da Silva et al., 2015*; *Morais et al., 2016*; *Rodrigues, Kalks & Tótola, 2015*; *Rodrigues et al., 2018*) using water and soil samples (Table 2, Fig. 2).

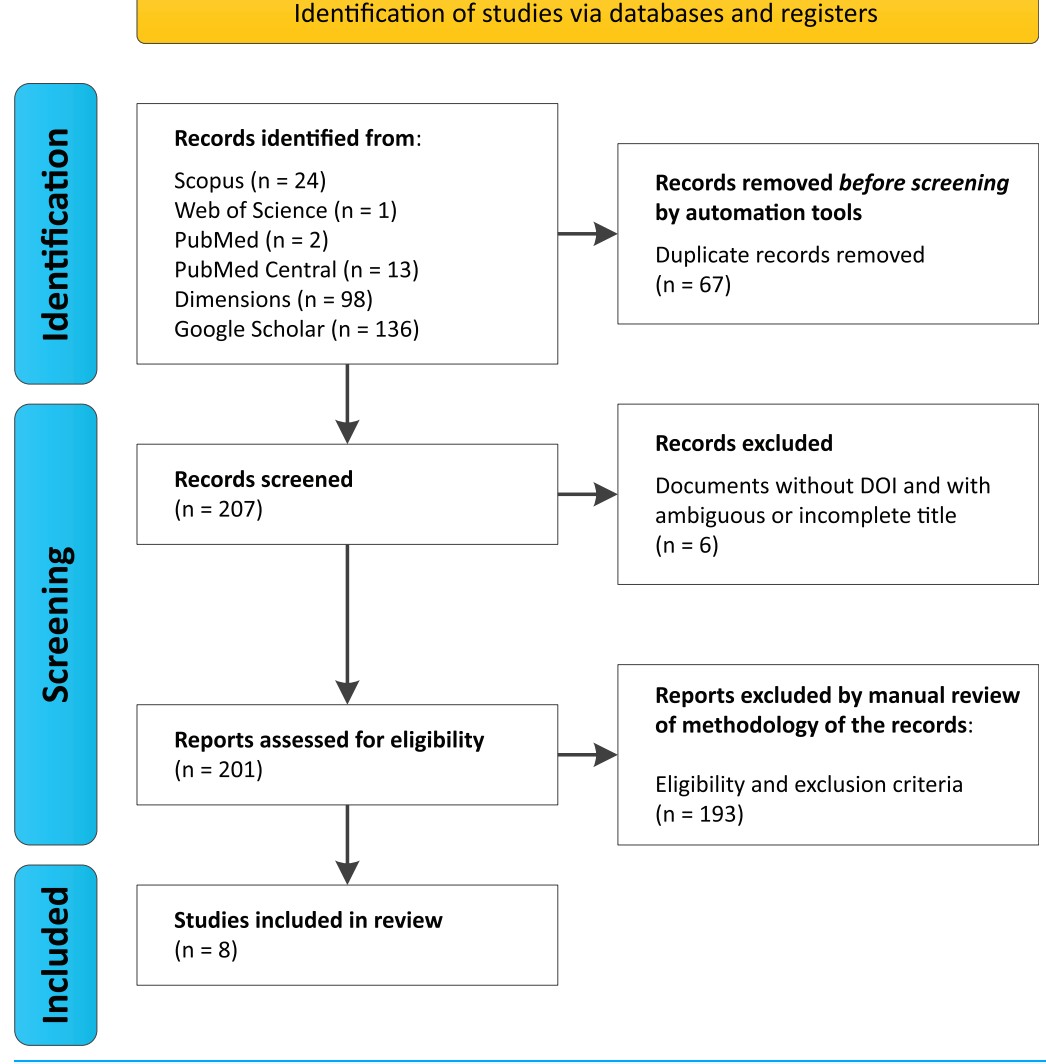

**Figure 1  PRISMA Flowchart for this systematic review.** We performed searches on August 2024 using six databases, considering the group of keywords "trindade island" and "metagenomics or shotgun or amplicons" and "16S or 18S or ITS or internal transcribed spacer".

For more detailed visualization of the collected sample locations, an interactive map is available: https://glenjasper.github.io/leaflet-trindade-island-review-map.

## Recovered communities of Archaea, Bacteria, and Fungi

This review focused on analyzing Archaea, Bacteria, and Fungi communities. Taxa from other kingdoms were also extracted but not analyzed; they can be consulted in Data S3. The selected articles reported results at varying taxonomic levels. *Meirelles et al. (2015)* and *Câmara et al. (2023)* presented results up to the genus level, while *Rodrigues et al. (2018)* reported data at the family level, and *Morais et al. (2016)* at the order level. *Rodrigues, Kalks & Tótola (2015)* and *da Silva et al. (2015)* used the SANGER method to sequence the

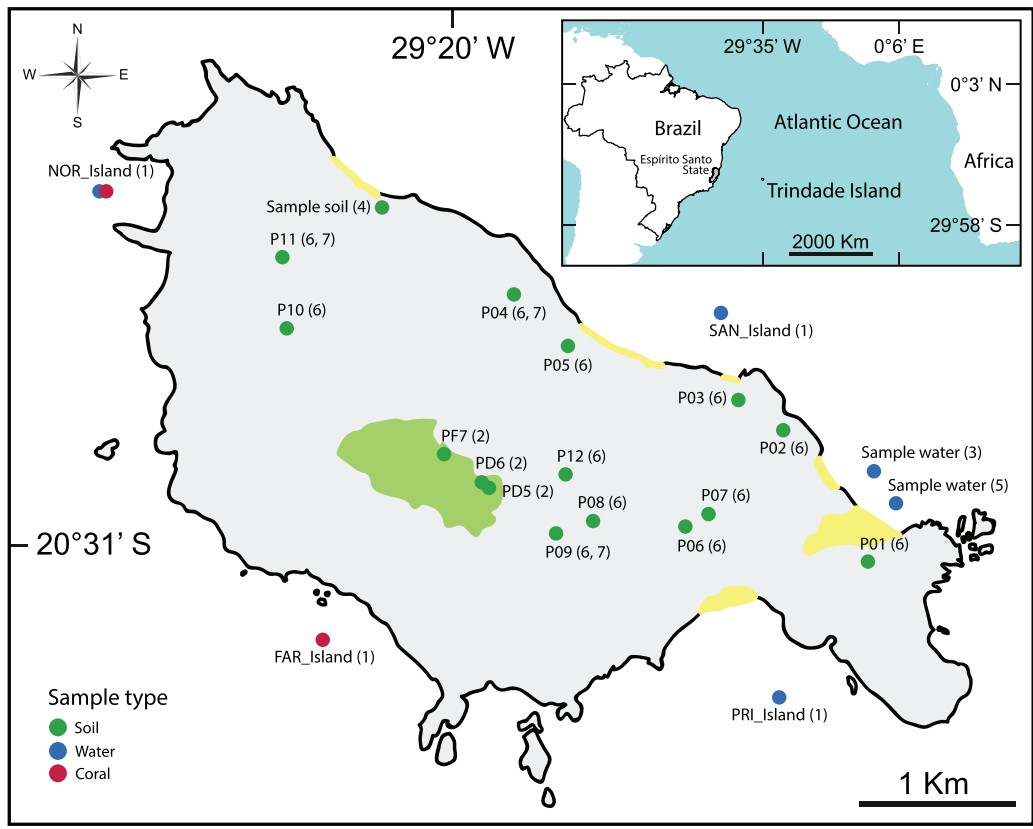

**Figure 2 Map showing the location of all types of samples analyzed in the selected articles.**
(1): *Meirelles et al. (2015)*, (2): *Câmara et al. (2023)*, (3): *Rodrigues et al. (2018)*, (4): *Morais et al. (2016)*, (5): *Rodrigues, Kalks & Tótola (2015)*, (6): *da Silva et al. (2015)*, (7): *Pylro et al. (2014)*.

16S rRNA gene, identifying bacterial communities restricted to a few species. As a result, the findings from these two studies were included only in the discussion.

Because of differences in the taxonomic data presented in the selected studies, comparative analyses were primarily conducted at the phylum, order, family, and genus levels. Although *Pylro et al. (2014)* met the selection criteria, the authors did not report taxonomic data, limiting their study to pipeline comparisons for amplicon analysis. Consequently, no relevant information was extracted, and this study was excluded from further analyses.

### Bacterial dominance and high rates of unclassified sequences in soil, with abundant low-frequency taxa in water and coral sample

In the analysis of Archaea and Bacteria, the majority of microbial reads annotated in the selected articles were identified as Bacteria, while Archaea were almost absent from the samples. No Archaea reads were detected in the microcosm samples, appearing only in the environmental samples. The highest relative abundances of Archaea were found in the coral samples NOR_Island_C (2.47%) and FAR_Island_C (5%) (Data S4).

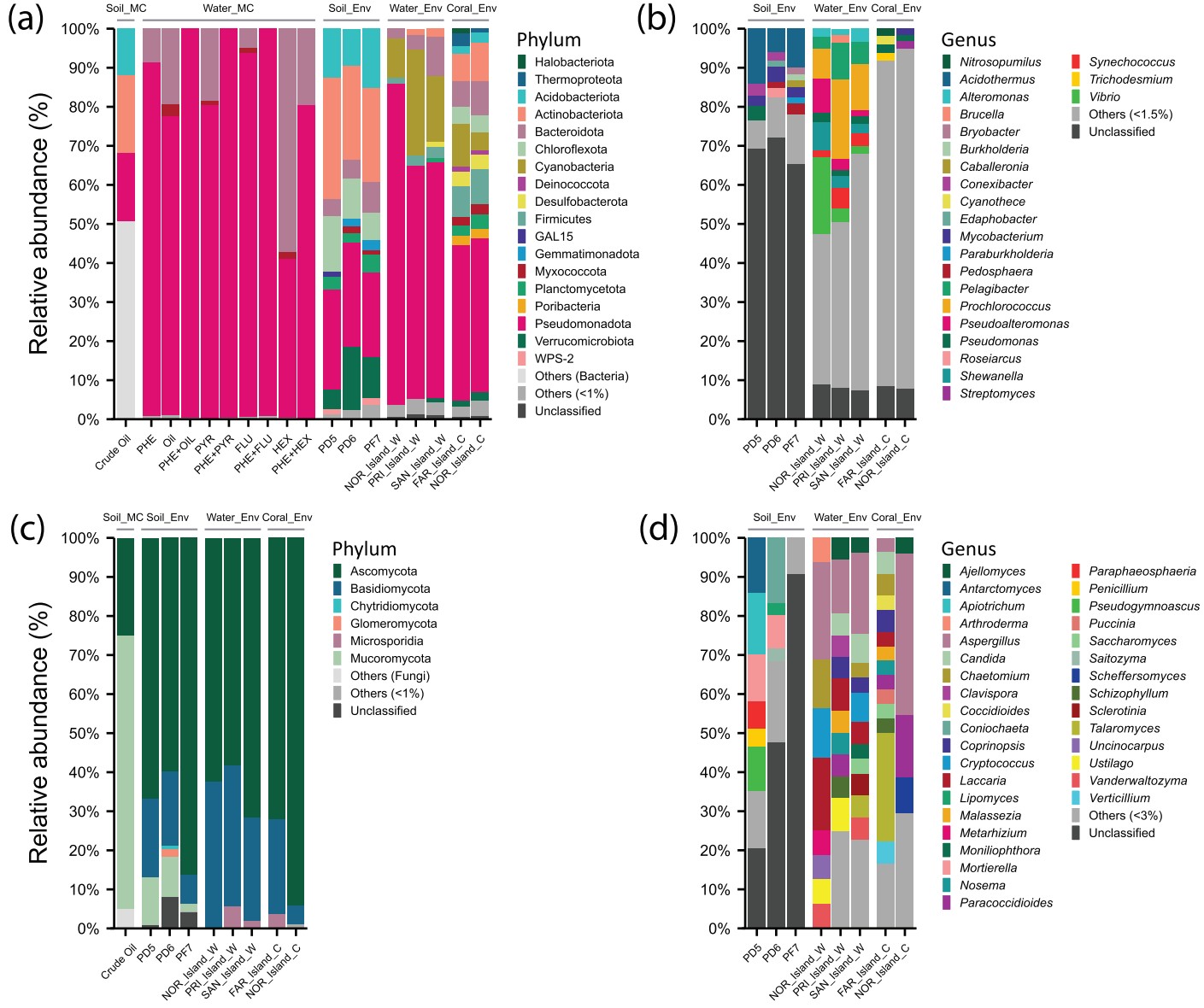

**Figure 3 Relative abundance of phyla and genera of Archaea, Bacteria, and Fungi in different environments: soil samples treated with microcosms (Soil_MC), twelve microcosm treatments using seawater samples (Water_MC), independent-cultivation soil samples (Soil_Env), independent-cultivation water samples (Water_Env), and independent-cultivation coral tissue samples (Coral_Env).** (A) Archaea and Bacteria phyla with relative abundance $\geq 1\%$. (B) Archaea and Bacteria genera with relative abundance $\geq 1.5\%$. (C) Fungal phyla with relative abundance $\geq 1\%$. (D) Fungal genera with relative abundance $\geq 3\%$. Taxa with relative abundances below the respective thresholds are grouped as "Others", while unclassified taxa are shown as "Unclassified".

A notable amount of unclassified reads was observed, particularly in environmental soil samples (PD5, PD6, PF7). At the family level, 36% of reads were unclassified, increasing to 68.9% at the genus level and 99.75% at the species level. However, one bacterial species, *Dinghuibacter silviterrae*, was identified in the Fazendinha region (PF7) (Data S4).

For the Water_Env and Coral_Env groups, about 8% of the reads were unclassified at the family and genus levels, decreasing to 1% at the species level (Data S4). This pattern

may be attributed to the presence of bacterial species labeled as "uncultured." However, a significant percentage of low-abundance taxa (<1%, <1.5%, or <3%) grouped as "Others" was observed in the Water_Env and Coral_Env samples at species, genus, family, and order levels (Data S4, Fig. 3B).

### Taxon distribution in soil samples

In environmental soil samples (PD5, PD6, PF7), only one Archaea class, Thermoplasmata (0.13%), was identified in low abundance, indicating a limited presence of this domain in the elevated regions of Trindade Island.

For Bacteria, 21 phyla and 34 genera were recorded. The phylum Actinobacteriota (26.31%) was prominent, with genera *Acidothermus* (10%), *Mycobacterium* (3%), and *Conexibacter* (2.11%). Within Pseudomonadota (24.68%), *Pseudomonas* (3.62%) was predominant in PD5 (Pico do Desejado), while *Roseiarcus* (2.38%) was more abundant in PD6, and *Pedomicrobium* (1.07%) appeared in smaller proportions. The phylum Acidobacteriota (12.52%) included *Bryobacter* (1.19%) and *Edaphobacter* (1.15%), with moderate distributions across all samples. *Verrucomicrobiota* (10.53%) was notable for Pedosphaera (1.65%), while *Chloroflexota* (10.50%) included Thermosporothrix (0.7%), identified exclusively in Fazendinha (PF7). The phylum Bacteroidota (5.75%) included low-abundance genera *Flavobacterium* (0.43%) and *Puia* (0.37%) (Figs. 3A and 3B, Data S4).

In Fungi, five phyla and 162 genera were identified. Ascomycota (70.1%) was dominant, with notable genera *Antarctomyces* (5.8%), *Coniochaeta* (5.57%), *Pseudogymnoascus* (4.79%), *Paraphaeosphaeria* (2.35%), *Penicillium* (2.01%), and *Lipomyces* (1.48%). Basidiomycota (15.5%) included *Apiotrichum* (5.90%) and *Saitozyma* (1.29%). In Mucoromycota (8.08%), *Mortierella* (7.44%), predominantly in Pico do Desejado, and *Umbelopsis* (0.34%) were identified. Glomeromycota (0.74%) and Chytridiomycota (0.36%) were present in small proportions (Figs. 3C and 3D, Data S4).

The site Fazendinha had the highest number of identified bacterial genera (30) compared to Pico do Desejado (25). Conversely, Pico do Desejado had more fungal genera (144) than Fazendinha (58) (Data S4).

In crude oil-treated soil samples (Crude Oil), no archaean taxa were identified. For Bacteria, Actinobacteriota (19.9%) was dominant, with Actinomycetales (17%) and Acidimicrobiales (2.9%). Pseudomonadota (17.3%) included orders such as Rhizobiales (6.4%), Burkholderiales (4%), Xanthomonadales (3.9%), and Sphingomonadales (3%). Acidobacteriota (11.9%) was primarily represented by Order iii1-15 (8.5%). Among fungi, there was a drastic reduction in diversity, with the dominance of Mortierellales (70%) from Mucoromycota, alongside Hypocreales (24%) and Botryosphaeriales (1.1%) from Ascomycota (Data S4).

### Taxon distribution in water samples

In independent-culturing seawater samples (NOR_Island_W, PRI_Island_W, SAN_Island_W), Archaea annotations were limited. Bacterial communities were dominated by Pseudomonadota (67.31%), with abundant genera including *Vibrio*

(19.54%), *Pseudoalteromonas* (8.78%), and *Shewanella* (7.06%) in NOR_Island_W, as well as *Pelagibacter* (6.11%), *Alteromonas* (2.43%), and *Pseudomonas* (2.07%) across all samples. Cyanobacteria (17.94%) was dominated by *Prochlorococcus* (13.24%) and *Synechococcus* (3.38%). Bacteroidota (5.48%) included genera *Bacteroides* (0.62%), *Polaribacter* (0.28%), and *Pedobacter* (0.28%) (Figs. 3A and 3B, Data S4).

In Fungi, Ascomycota (64.18%) dominated, with notable genera *Aspergillus* (18.88%), *Chaetomium* (5.42%), *Candida* (4.37%), and *Vanderwaltozyma* (3.97%). Basidiomycota (33.34%) included *Laccaria* (10.91%), *Cryptococcus* (7.60%), and *Ustilago* (4.86%). PRI_Island (20) and SAN_Island (24) exhibited greater fungal genus diversity than NOR_Island (9). However, bacterial genus abundance was more uniform among the samples (Figs. 3C and 3D, Data S4).

In the Water_MC group, consisting of nine hydrocarbon-treated samples (*e.g.*, PHE, OIL, PYR, *etc.*), no archaean taxa were identified. Bacteria were dominated by Pseudomonadota (84.38%), represented by families Oceanospirillaceae (34.83%), Alcanivoracaceae (19.31%), Rhodobacteraceae (11.59%), Sphingomonadaceae (4.01%), and Hyphomicrobiaceae (3.63%). Moreover, Bacteroidota (14.45%) included families Flavobacteriaceae (11.45%) and Cryomorphaceae (2.08%) (Data S4).

### Taxon distribution in coral tissue samples

In coral tissue samples (FAR_Island_C, NOR_Island_C), Archaea were moderately represented, primarily by the phylum Thermoproteota (3.49%) with the genus *Nitrosopumilus* (1.81%), and the phylum Halobacteriota (1.14%) with the genus *Methanosarcina* (0.13%).

In the Bacteria domain, dominant taxa included: phylum Pseudomonadota (39.7%), with genera *Pseudomonas* (1.86%) and *Burkholderia* (1.28%); phylum Actinobacteriota (8.53%) with *Streptomyces* (1.44%) and *Mycobacterium* (1.41%), phylum Firmicutes (8.39%) with genera *Bacillus* (1.06%) and *Clostridium* (0.99%), Cyanobacteria (7.78%) with *Cyanothece* (1.75%) and *Trichodesmium* (1.06%). The phylum Bacteroidota (7.55%) was best represented by the genera *Bacteroides* (0.80%), and *Rhodothermus* (0.66%). The phylum Chloroflexota (4.42%) was represented by the genera *Roseiflexus* (1.09%) and *Chloroflexus* (0.87%) (Figs. 3A and 3B, Data S4).

In the Kingdom Fungi, the dominant taxa were from the phylum Ascomycota (83.15%), with notable genera including *Aspergillus* (41.34%), *Paracoccidioides* (16.14%) and *Scheffers* (9.06%). In the NOR_Island_C sample, the predominant genus was *Pichia* (9.06%), while in the FAR_Island_C sample, the main genera were *Talaromyces* (27.78%) and *Chaetomium* (5.56%).

The phylum Basidiomycota (14.40%) included *Schizophyllum* (3.23%) and *Malassezia* (2.44%) in both samples, with *Coprinopsis* (5.56%) exclusively in FAR_Island_C. Additionally, the phylum Microsporidia (2.05%) was represented solely by the genus *Nosema* (2.05%) (Figs. 3C and 3D, Data S4). At Ponta Noroeste, a higher diversity of fungal genera (33) was detected compared to Ponta dos Cinco Farilhões, where 23 genera were identified. This trend was also observed for bacterial genera, with Ponta Noroeste

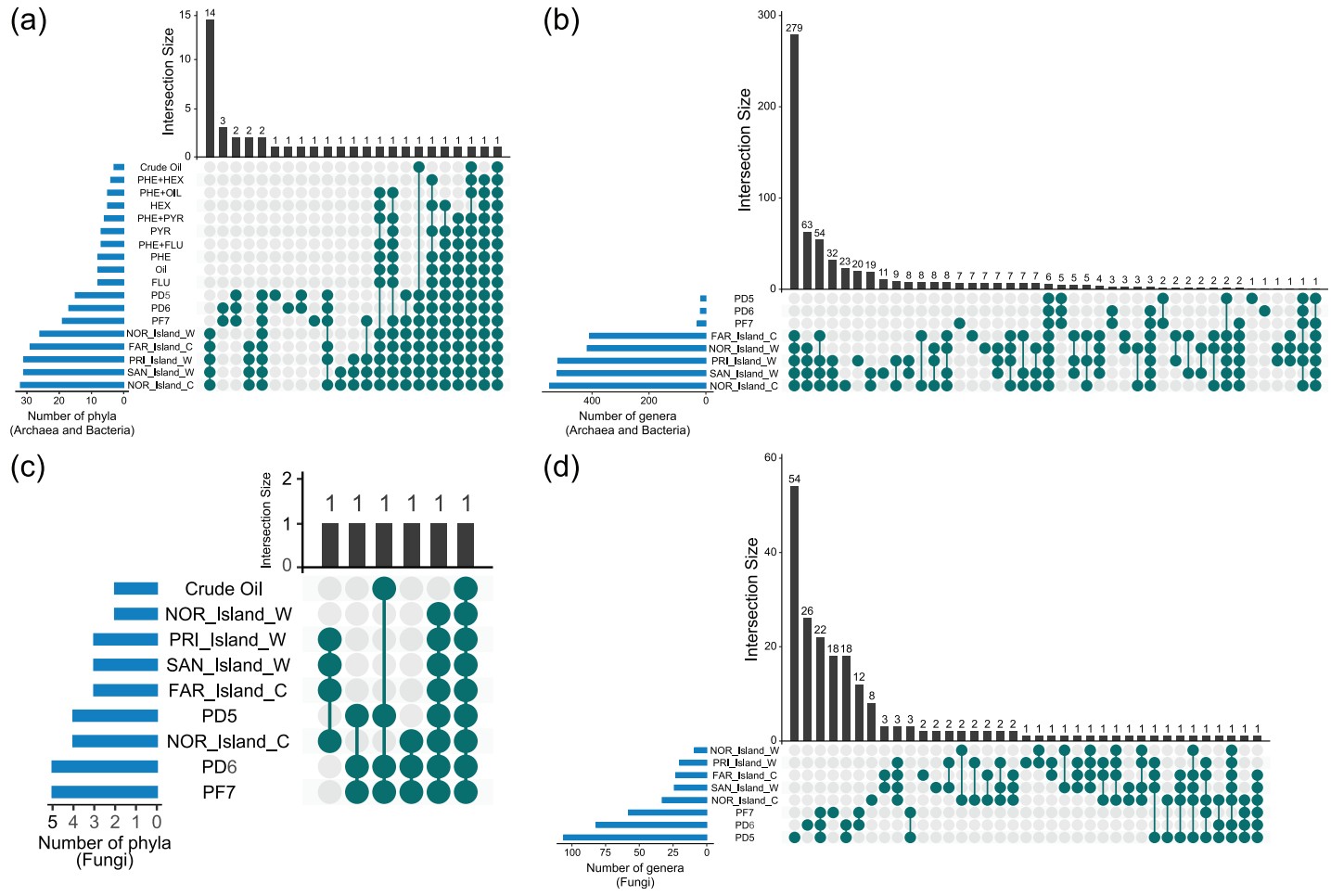

**Figure 4 Sharing of phyla (A, C) and genera (B, D) of Archaea, Bacteria and Fungi between culture-independent samples and microcosm-treated samples.**

showing greater diversity (550 genera) compared to Ponta dos Cinco Farilhões (409 genera) (Data S4).

### Taxon sharing among samples

UpSet diagram analyses revealed shared and unique taxa among Archaea, Bacteria, and Fungi across environmental samples (Soil_Env, Water_Env, and Coral_Env) and microcosm-treated samples (Soil_MC and Water_MC). The focus was on taxonomic categories at the phylum and genus levels (Fig. 4, Data S5).

Based on the data analyzed, 26 phyla of Bacteria and Archaea were present in both the environmental water and coral tissue samples. Among those samples, 14 were unique to the respective environments, namely Aquificota, Campylobacterota, Chrysiogenota, Deferribacterota, Deinococcota, Desulfobacterota, Fusobacteriota, Halobacteriota, Methanobacteriota, Poribacteria, Spirochaetota, Synergistota, Thermoproteota and Thermotogota. A total of 12 bacterial phyla were identified in the environmental soil

samples, with two (GAL15 and WPS-2) being exclusive. In the samples treated with microcosms, only two bacterial phyla (Bacteroidota and Pseudomonadota) were identified, with the Pseudomonadota phylum being the sole presence across all the groups of samples analyzed. It is noteworthy that the highest diversity of phyla was observed in the environmental water and coral tissue samples, while the soil and water samples treated with microcosms exhibited the lowest microbial richness (Fig. 4A, Data S5).

Fungal phyla were more prevalent in soil environmental samples, followed by water and coral tissue samples. Only two fungal phyla (Ascomycota and Mucoromycota) were found in microcosm-treated samples. Ascomycota was the only phylum identified across all the fungal sample groups (Fig. 4C, Data S5).

At the genus level, 293 genera of Archaea and Bacteria were shared between water and coral tissue environments, with 279 unique to these environments. Seven genera (*Baumannia*, *Buchnera*, *Carnobacterium*, *Chlamydia*, *Mobiluncus*, *Riemerella* and *Simonsiella*) were exclusively shared by seawater samples, while eight (*Brachybacterium*, *Brevibacterium*, *Halorubrum*, *Haloterrigena*, *Methanocorpusculum*, *Oribacterium*, *Pyrobaculum* and *Tolypothrix*) were unique to coral tissue. Soil samples shared 12 bacterial genera, with five unique to this environment (*Bryobacter*, *Edaphobacter*, *Gemmatimonas*, *Pedomicrobium*, and *Roseiarcus*). The Fazendinha (PF7) region had seven unique genera (*Chthonomonas*, *Dinghuibacter*, *Edaphobaculum*, *Inquilinus*, *Occallatibacter*, *Roseisolibacter*, and *Thermosporothrix*). Only six genera (*Conexibacter*, *Haliangium*, *Mucilaginibacter*, *Mycobacterium*, *Pedosphaera*, and *Rhodomicrobium*) were shared across all sample groups. Bacterial genus diversity was significantly higher in marine regions compared to soil areas (Fig. 4B, Data S5).

No fungal genus was shared across all samples. Coral tissue samples had two unique genera (*Grosmannia* e *Rhodonia*). Soil samples shared 22 unique genera, including *Alternaria*, *Antarctomyces*, *Apiotrichum*, *Archaeorhizomyces*, *Ciliophora*, *Cladosporium*, *Diaporthe*, *Hyphodiscus*, *Juncaceicola*, *Knufia*, *Lipomyces*, *Mortierella*, *Neoascochyta*, *Papiliotrema*, *Penicillium*, *Pestalotiopsis*, *Pseudogymnoascus*, *Purpureocillium*, *Recurvomyces*, *Rhexocercosporidium*, *Saitozyma*, and *Trichoderma*. The Fazendinha (PF7) region displayed 18 unique genera (*Acaulospora*, *Arachnomyces*, *Botrytis*, *Callistosporium*, *Chaetosphaeria*, *Circinella*, *Clavaria*, *Cryptodiscus*, *Gloeocystidiellum*, *Gloioxanthomyces*, *Hyphozyma*, *Mycena*, *Phaeotremella*, *Podospora*, *Solicoccozyma*, *Spencermartinsiella*, *Trechispora*, and *Triparticalcar*), while the Pico do Desejado (PD6) region also had 18 unique genera (*Acremonium*, *Arthrinium*, *Debaryomyces*, *Gibberella*, *Glarea*, *Gremmenia*, *Hyaloscypha*, *Ilyonectria*, *Leucosporidiella*, *Macrophomina*, *Pezicula*, *Pezoloma*, *Phenoliferia*, *Rhodotorula*, *Thelebolus*, *Trichosporon*, *Veronaeopsis*, and *Volutella*). Unlike bacterial taxa, fungal genera exhibited greater diversity in soil samples compared to marine samples (Fig. 4D, Data S5).

### Analysis of alpha and beta diversity and sample correlation

Alpha diversity analysis revealed distinct results across sample groups. Water_MC group showed a mean Shannon index of 1.109, indicating relatively low diversity in this specific environment. Soil_MC group exhibited a mean Shannon index of 1.683, reflecting

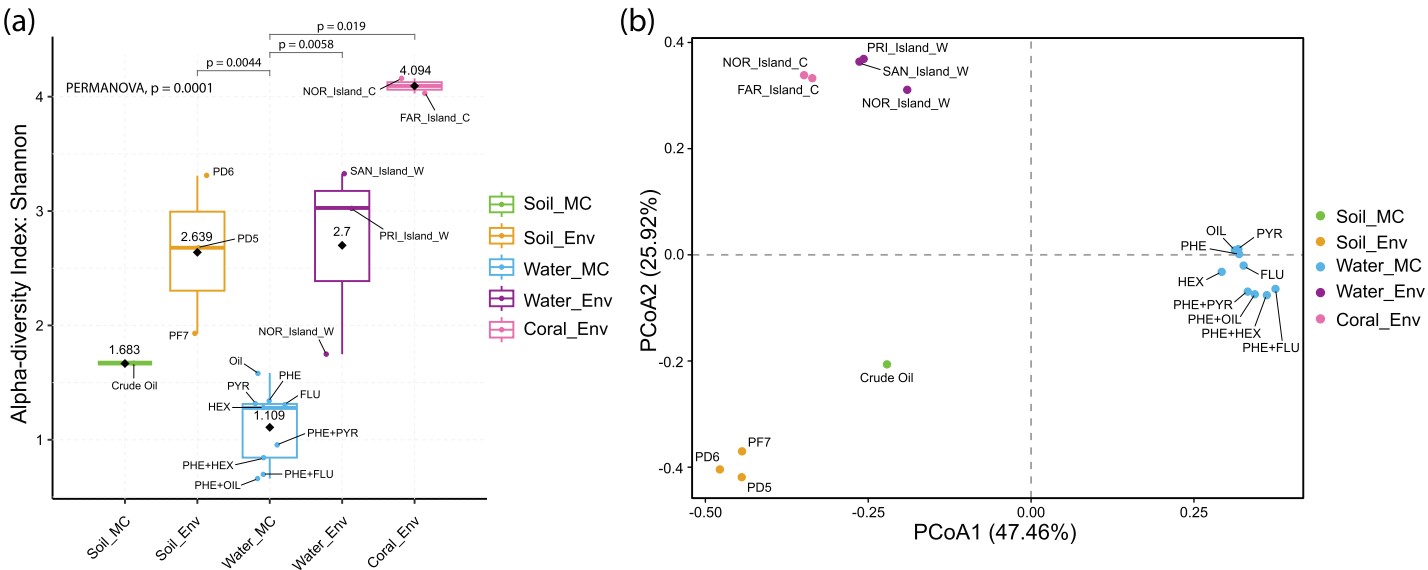

**Figure 5** **Alpha (A) and beta (B) diversity of culture-independent and microcosm-treated samples at the taxonomic Order level.** The alpha diversity are corroborated by the significant differences found between groups using the PERMANOVA analysis.

moderate diversity. Soil_Env and Water_Env groups displayed significantly higher Shannon indices, 2.639 and 2.7, respectively, indicating substantial diversity compared to microcosm treated samples. Within Soil_Env, samples from Pico do Desejado (2.68 and 3.31) demonstrated greater diversity than those from Fazendinha (1.93). Coral_Env group had the highest mean Shannon index, 4.094, highlighting a notably rich biodiversity in independent-cultivation coral tissue samples (Fig. 5A).

Beta diversity, represented by Principal Coordinate Analysis (PCoA) (Fig. 5B), revealed clustering of samples based on similarity: Water_Env and Coral_Env groups were closely positioned, showing no statistically significant differences according to PERMANOVA analyses (Data S6). Only Water_MC samples exhibited significant differences when compared to Soil_Env ($p = 0.0044$), Water_Env ($p = 0.0058$), and Coral_Env ($p = 0.019$).

Strong association networks at the phylum level showed that Coral_Env had the highest number of associated phyla, with no significant associations in only five phyla (Fig. 6A). Coral samples shared more associations with Water_Env samples than with soil samples (Soil_Env and Soil_MC), consistent with PERMANOVA results. Unexpectedly, Soil_Env and Soil_MC groups exhibited limited overlap in associations, with Soil_Env showing more connections with Coral_Env samples.

Water_MC and Soil_MC groups displayed fewer associations: soil-associated phyla included Ascomycota, Actinobacteriota, and Acidobacteriota, while water-associated phyla included only Pseudomonadota. Across all substrates, bacterial phyla dominated associations, while fungal phyla (Ascomycota and Basidiomycota) were associated only with Soil_Env and Soil_MC, respectively. Methanobacteriota and Halobacteriota, representing Archaea, were associated exclusively with Coral_Env. Despite their high abundance (Fig. 6A), Ascomycota and Pseudomonadota were not

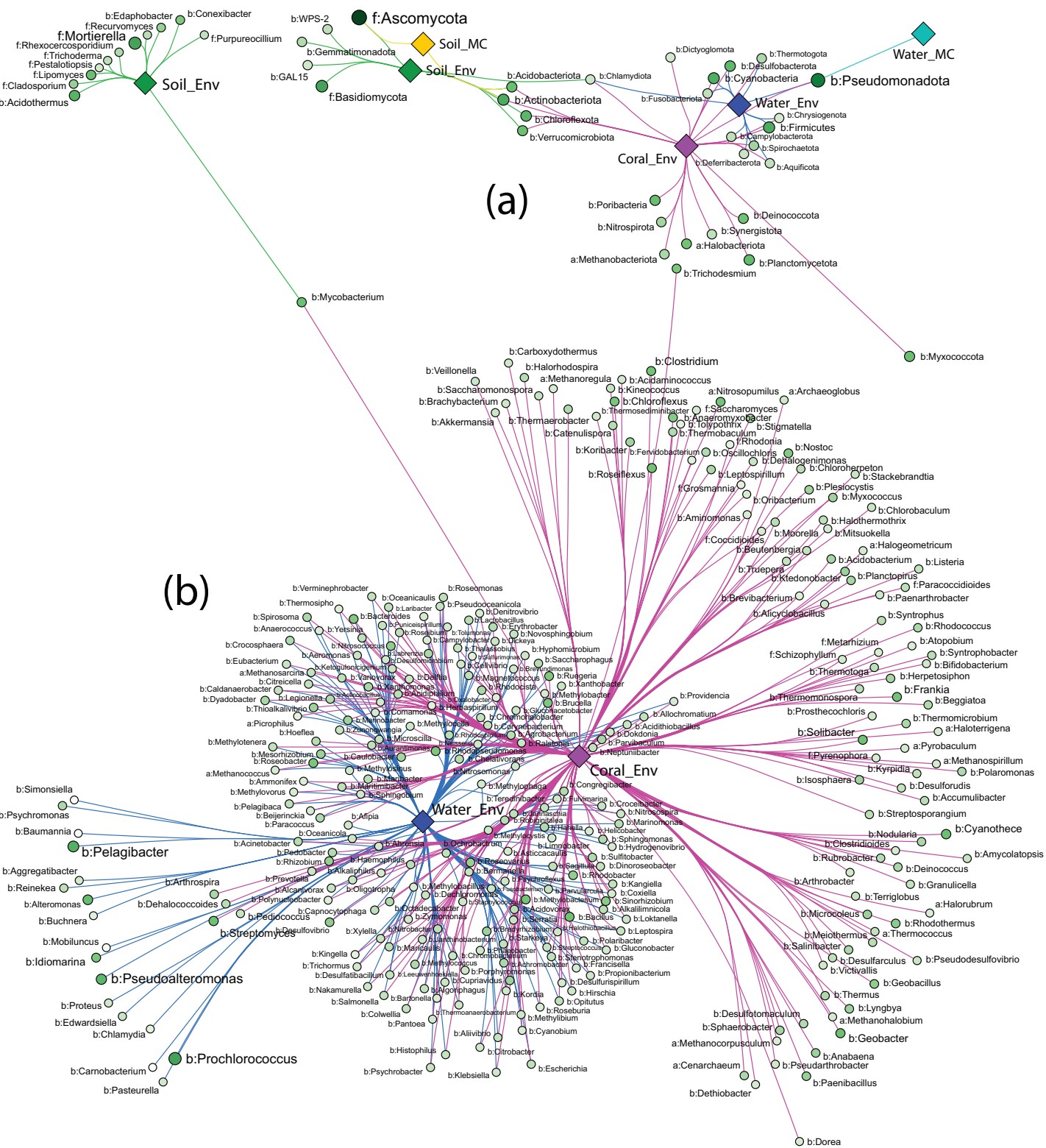

**Figure 6 Network of strong associations between the phyla (A) and genera (B) of Archaea, Bacteria, and Fungi with the sample groups.**

significantly associated with all substrates. Actinobacteriota, Chlamydiota, and Pseudomonadota showed the highest number of associations, linked to three out of five substrate types.

At the order level, the association patterns mirrored those at the phylum level, with Coral_Env displaying the majority of the associations. Microcosm-treated samples exhibited fewer associations across orders, while Coral_Env and Water_Env showed a substantial number of shared associations. Soil samples demonstrated no overlap in their associations (Fig. S1).

At the genus level, only environmental samples (Soil_Env, Water_Env, and Coral_Env) identified communities of Archaea, Bacteria, and Fungi. Coral tissue samples exhibited the highest number of significantly associated genera, including most archaeal genera, followed by water samples. Shared associations between water and coral tissue samples were predominantly bacterial, with fewer archaeal and fungal genera represented. In independent-culturing seawater samples, the most abundant bacterial genera were *Prochlorococcus*, *Pelagibacter*, *Pseudoalteromonas*, and *Alteromonas*, while in coral tissue samples, the most abundant were *Cyanothece*, *Streptomyces*, *Mycobacterium*, and *Geobacter* (Fig. 6B).

Soil samples showed fewer strong associations, with bacterial genera *Acidothermus* and *Mycobacterium* as those of the most abundant. Fungal genera *Mortierella* and *Lipomyces* exhibited high abundance and strong associations with samples from Pico do Desejado and Fazendinha (Fig. 6B).

### Biotechnological potential of bacterial isolates identified by partial sequencing of the 16S rRNA gene

Two additional studies utilized the SANGER sequencing method for partial sequencing of the 16S rRNA gene from bacterial isolates. Because of that methodological specificity, only a few bacterial taxa were identified, precluding their inclusion in statistical analyses. Nonetheless, these taxa are noteworthy for their biotechnological potential. *Rodrigues, Kalks & Tótola (2015)* identified bacterial strains capable of degrading various hydrocarbons. Using isolates from water samples, species such as *Rhodococcus rhodochrous* and *Nocardia farcinica* demonstrated the ability to degrade a wide range of hydrocarbons, including aliphatic, aromatic, and PAH compounds, such as toluene, octane, xylene, naphthalene, phenanthrene, pyrene, hexadecane, anthracene, eicosane, tetracosane, triacontane, and pentacontane. Additional species included *Cellulosimicrobium cellulans* (hexadecane degradation) and *Microbacterium lacticum* (naphthalene degradation). Strains from the genera *Exiguobacterium*, *Microbacterium*, and *Tistrella* exhibited selective degradation capabilities for hydrocarbons such as anthracene, hexadecane, naphthalene, tetracosane, octane, phenanthrene, pyrene, and xylene. *da Silva et al. (2015)* identified strains from the genus *Bacillus*, isolated from soil samples, capable of producing biosurfactants under high salinity conditions. These findings highlight the relevance of these bacterial isolates for bioremediation and industrial applications.

## DISCUSSION

This review focused on six studies that employed metabarcoding and metagenomics to characterize the microbial communities on Trindade Island. Although eight articles were initially selected, *Câmara et al. (2022)* was excluded because it presented the same results as *Câmara et al. (2023)*. Likewise, the study by *Pylro et al. (2014)* was excluded because it primarily focused on evaluating bioinformatics pipelines without providing diversity data for the sequenced soil samples. Of the six remaining studies, two analyzed uncultured environmental samples (*Câmara et al., 2023*; *Meirelles et al., 2015*), two utilized treated samples in microcosms (*da Silva et al., 2015*; *Morais et al., 2016*), and two were based on cultivated samples (*Rodrigues, Kalks & Tótola, 2015*; *Rodrigues et al., 2018*). Five studies employed marker gene sequencing (16S rRNA and/or ITS) (*Câmara et al., 2023*; *da Silva et al., 2015*; *Morais et al., 2016*; *Rodrigues, Kalks & Tótola, 2015*; *Rodrigues et al., 2018*), whereas only one used shotgun metagenomic sequencing (*Meirelles et al., 2015*). The most commonly used sequencing platform was Illumina MiSeq, followed by 454 GS FLX Titanium, Ion Torrent, and, lastly, the Sanger method.

Despite the limited molecular data on Trindade Island's microbial diversity, *Costa-Rezende et al. (2023)* conducted a comprehensive review of over 100 years of scientific research on the island, identifying 312 references primarily in the fields of biology, health sciences, and agriculture. Their findings underscore that microbial investigation has been significantly underestimated, noting that the first DNA-based study was only published in 2014 (*Pylro et al., 2014*).

The only study employing a shotgun metagenomic approach used 454 pyrosequencing, a technology that, although innovative at the time, has limitations compared to current methods. Consequently, there is a substantial opportunity to refine environmental metagenomic studies using more advanced platforms, such as Illumina short-read sequencing or Oxford Nanopore Technologies (ONT) long-read sequencing. These platforms can offer more detailed and comprehensive insights into the microbial and fungal diversity of the island.

In addition to studies investigating environmental DNA, only three genomic studies have been published on Trindade Island, each describing specific bacterial genomes. These include *Rhodococcus rhodochrous* TRN7 (*Rodrigues et al., 2016*), *Nocardia farcinica* TRH1 (*Rodrigues et al., 2017*), and *Staphylococcus warneri* TRPF4 (*Freitas et al., 2020*). Collectively, these studies highlight the need for further DNA-based research to explore the island's microbial diversity.

### Bacterial and fungal communities in the ecological niches of Trindade Island marine environments

*Meirelles et al. (2015)* provided a comprehensive overview of microbial communities in the marine areas of Trindade Island, revealing notable differences among habitats. The Trindade Shelf region (SAN_Island)—which includes the Jaseur Seamount, Columbia Bank, Almirante Saldanha Seamount, Vitória Seamount, and Eclaireur Seamount, all part of the Vitória-Trindade Chain (VTC)—displayed unique microbiota, especially in terms of

bacterial diversity. These sites, which featured rhodolith beds overgrown by fleshy algae, exhibited both high abundance and diversity of microbial communities in the water column.

The study reported the highest concentrations of chlorophyll $a$ (0.18–0.27 µg L$^{-1}$) and orthophosphate (0.55 µM) in these areas, suggesting higher primary productivity and potential benthic–pelagic coupling. These conditions presumably favor more diverse bacterial communities because of the consistent input of nutrients from fleshy algae and the accumulation of dissolved organic carbon (DOC) released by these algae.

In the Trindade Shelf region, the phylum Pseudomonadota was dominant, playing an especially significant role in heterotrophic processes and nutrient recycling. The phyla Bacteroidota, Actinobacteriota, and Verrucomicrobiota were also prominent, performing key functions in organic compound degradation and carbon cycling—processes essential for sustaining high productivity on the Trindade Shelf.

*Meirelles et al. (2015)* also investigated three other sites on Trindade Island: Ponta dos Cinco Farilhões (FAR_Island), Ponta Noroeste (NOR_Island), and Enseada do Príncipe (PRI_Island). These sites featured rocky reefs predominantly covered by turf algae, with higher DOC concentrations (up to 219.17 µM at Ponta dos Cinco Farilhões). The associated microbiota appeared adapted to these environmental conditions. Turf algae, which extensively cover these reefs, release exudates rich in organic compounds, potentially promoting bacterial growth while also creating local hypoxia that can stress corals. Cyanobacteria, such as *Lyngbya*, are often integral to turf algae in these environments, releasing DOC and exacerbating unfavorable conditions for corals.

Bacteroidota and Pseudomonadota were the dominant bacterial phyla in these reefs. Bacteroidota, known for their ability to decompose organic matter, and Pseudomonadota, typically fast-growing heterotrophic bacteria found in DOC-rich environments, both reflect the substantial influence of turf algae and cyanobacteria. Elevated irradiance and organic input may further drive microbial activity, potentially worsening coral health.

*Meirelles et al. (2015)* observed that *Mussismilia hispida* colonies in these locations showed impaired health, including tissue necrosis and bleaching in over 90% of colonies in shallow reefs. The increased presence of Bacteroidota and Ascomycota in these corals reinforces their possible links to stress and disease, as both phyla have been reported in diseased corals in other reef systems (*Schmieder & Edwards, 2011*). Although the authors did not examine fungal communities in detail, the broad taxonomic composition supports differences in bacterial and fungal assemblages between deeper areas (SAN_Island) and shallow reefs (FAR_Island, NOR_Island, PRI_Island). The deeper Trindade Shelf region displays higher bacterial diversity, correlating with elevated nutrient levels and diminished light stress, whereas the shallow, turf algae–dominated reefs show bacterial communities adapted to high DOC levels and low-oxygen conditions, adversely affecting coral health.

## Terrestrial environments

*Câmara et al. (2023)* investigated three soil samples from two high-elevation sites on the island: Pico do Desejado (PD5 and PD6) and Fazendinha (PF7). These soils were characterized by low base saturation (PBS < 50%), high cation exchange capacity (CEC),

and high organic matter content, mainly composed of plant fibers. The presence of iron from weathered ferromagnesian minerals fosters microbial communities adapted to acidic, metal-rich conditions. Such soil properties significantly influence both the structure and function of bacterial and fungal communities.

In this study, the dominant bacterial phyla were Actinobacteriota, Pseudomonadota, Acidobacteriota, Chloroflexota, and Verrucomicrobiota, all typical of acidic soils with high organic matter. Actinobacteriota were most abundant at Pico do Desejado (PD5), reflecting their crucial role in decomposing complex organic matter (*e.g.*, lignin and cellulose) and driving carbon cycling. Their abundance may be linked to the active breakdown of plant residues, such as fibers from giant ferns (*Cyathea gigantea*), which predominate in the Fazendinha region. By contrast, PD6 was dominated by Pseudomonadota and Verrucomicrobiota, which are pivotal in the nitrogen cycle, including nitrification and nitrogen fixation. Their prevalence could be explained by the buildup of nutrients due to the weathering of the island's rocky substrate.

At Fazendinha (PF7), Acidobacteriota and Bacteroidota were the most prominent phyla. Acidobacteriota, commonly found in acidic environments, drive essential decomposition and mineralization processes in nutrient-poor soils. The high potential acidity (H+Al) and low pH in Fazendinha soils account for the predominance of this phylum, critical for maintaining fertility under challenging conditions. Bacteroidota further support nutrient cycling by breaking down organic matter.

Fungal communities in these soils were also highly diverse, with Ascomycota, Basidiomycota, and Mucoromycota as the main phyla (*Câmara et al., 2023*). Notably, *Antarctomyces psychrotrophicus* and *Pseudogymnoascus* sp. exhibited adaptations to cold conditions, assisting in carbon recycling at low temperatures. *Mortierella humilis* was particularly abundant at Fazendinha (PF7), highlighting its capacity to decompose complex organic compounds and form mycorrhizal associations that enhance phosphorus uptake by plants. These mycorrhizal interactions are instrumental in soil structuring and improve both stability and water retention.

Overall, these microbe–soil interactions emphasize the resilience of Trindade Island's ecosystems, even under adverse environmental conditions, and underscore the importance of microbial communities in maintaining the ecological balance of island habitats.

## Bacterial and fungal communities from microcosm-treated samples

*Rodrigues et al. (2018)* conducted microcosm experiments by enriching seawater with various hydrocarbons to evaluate their effects on microbial communities. The nine experimental treatments—PHE (phenanthrene), OIL (weathered oil), PHE+OIL (phenanthrene + weathered oil), PYR (pyrene), PHE+PYR (phenanthrene + pyrene), FLU (fluoranthene), PHE+FLU (phenanthrene + fluoranthene), HEX (hexadecane), and PHE +HEX (phenanthrene + hexadecane)—had distinct impacts on both the structure and function of the microbial communities. These differences reflect how microorganisms adapt to the presence of aliphatic *versus* aromatic hydrocarbons and their mixtures.

According to *Rodrigues et al. (2018)*, alpha and beta diversity differed markedly among treatments. Taxonomic shifts underscored how the chemical complexity of hydrocarbons

influences the adaptation and metabolic activity of marine microorganisms. Treatments containing only aromatic hydrocarbons (PHE, PYR, and FLU) significantly decreased microbial diversity, likely due to the inherent toxicity of polycyclic aromatic hydrocarbons (PAHs). Phenanthrene (PHE) promoted the dominance of *Alteromonas*, indicating the genus's specialized ability to degrade recalcitrant compounds. Pyrene (PYR), being highly hydrophobic, resulted in a more selective community dominated by taxa such as Oceanospirillaceae. Fluoranthene (FLU) had a subtler effect on beta diversity, with *Alteromonas* maintaining a strong presence but *Flavobacteriaceae* showing sensitivity, suggesting difficulties in metabolizing isolated PAHs.

Treatments combining phenanthrene with other PAHs (PHE+PYR and PHE+FLU) exacerbated the toxic effects, further reducing microbial diversity. In PHE+PYR, the family Sphingomonadaceae was abundant, suggesting a competitive advantage under PAH-rich conditions. In PHE+FLU, *Alteromonas* again emerged as dominant, highlighting its importance in degrading recalcitrant compounds. However, the reduced abundance of Cryomorphaceae implies that certain families struggle to adapt to multiple toxic compounds.

Hexadecane (HEX), an aliphatic hydrocarbon, supported the highest microbial diversity among the treatments. This less recalcitrant compound enabled a balance of Flavobacteria, Gammaproteobacteria, and Alphaproteobacteria, with Flavobacteriaceae being the most prevalent family. The combination of phenanthrene and hexadecane (PHE+HEX) displayed significant synergy, as hexadecane acted as a non-aqueous phase liquid (NAPL), enhancing the bioavailability and degradation of phenanthrene. This treatment stimulated the growth of *Alcanivorax* and *Alteromonas*, highlighting a complementary metabolic interaction between these genera.

In treatments involving weathered oil (OIL and PHE+OIL), *Alcanivorax* predominated, reflecting its specialization in degrading complex hydrocarbons and long-chain alkanes. Weathered oil, containing a diverse array of organic compounds, promoted increased abundance of taxa such as *Alteromonas* and *Marinomonas*, which are adapted to polluted environments. In the PHE+OIL treatment, *Oceanospirillaceae* and Flavobacteriaceae likewise increased, suggesting metabolic interactions that facilitate PAH degradation in the presence of alkanes.

Across these treatments, *Alteromonas*, *Alcanivorax*, and the Flavobacteriaceae family proved central to hydrocarbon degradation, each responding differently to aliphatic and aromatic compounds. While *Alteromonas* was particularly prevalent in PAH treatments, *Alcanivorax* was favored by alkanes and weathered oil. Overall, *Rodrigues et al. (2018)* concluded that interactions among various hydrocarbons modulate microbial metabolism and community composition, enhancing degradation efficiency under mixed contamination scenarios.

*Morais et al. (2016)* performed microcosm experiments with soil to assess how crude oil influences microbial and fungal communities on Trindade Island. Oil addition altered community structure, diversity, and composition, emphasizing an adaptive response to complex hydrocarbons. Microbial alpha diversity decreased in the presence of oil, with a reduction of up to 40% in fungal taxa—likely due to PAH toxicity, which eliminates

less-adapted organisms. Taxonomic analysis revealed Actinobacteriota, especially *Nocardia* and *Streptomyces*, as dominant in contaminated soil. These genera are known to produce biosurfactants and extracellular enzymes that enhance hydrocarbon bioavailability and promote degradation (*Balachandran et al., 2012*; *Rodrigues, Kalks & Tótola, 2015*).

Among Pseudomonadota, Betaproteobacteria and Gammaproteobacteria showed relative increases; this included the family Comamonadaceae, noted for its heterotrophic denitrification capabilities. In contrast, Alphaproteobacteria and Deltaproteobacteria, such as Rhodospirillales and Syntrophobacteraceae, decreased, indicating lower resilience to hydrocarbon exposure.

In the fungal community, *Mortierella* accounted for 70% of relative abundance under oil treatment. This oleaginous fungus metabolizes hydrocarbons (*Hughes, Bridge & Clark, 2007*) and accumulates lipids, making it central to soil adaptation in contaminated environments. Genera such as Hypocreales declined, reflecting limited capacity to endure adverse conditions.

## Bacterial communities from cultivated samples

*Rodrigues, Kalks & Tótola (2015)* isolated and characterized hydrocarbon-degrading microorganisms by collecting coastal sediment samples from Trindade Island. Fifteen bacterial isolates were identified, spanning the phyla Actinobacteriota, Firmicutes, and Pseudomonadota. Of these, the genera *Rhodococcus* and *Nocardia* showed remarkable catabolic versatility, capable of utilizing a variety of hydrocarbons—toluene, octane, xylene, naphthalene, phenanthrene, pyrene, hexadecane, anthracene, eicosane, tetracosane, triacontane, and pentacosane—as carbon and energy sources. In particular, *Rhodococcus rhodochrous*, which harbors the *alkB* and *C23DO* genes, exhibited rapid growth on both long-chain hydrocarbons and polycyclic aromatic hydrocarbons (PAHs). *Nocardia farcinica*, containing the *C23DO* gene, proved specialized in degrading specific aromatic hydrocarbons.

Although *Cellulosimicrobium cellulans* and *Microbacterium lacticum* were capable of degrading hexadecane and naphthalene, respectively, the authors recommend further studies to clarify the specific roles of these relatively understudied isolates. They conclude that the hydrocarbon-degrading capacity of these strains—isolated from an environment without a documented history of anthropogenic pollution—underscores the potential of indigenous microbiota for bioremediation, particularly given the island's proximity to Brazil's oil-producing regions.

*da Silva et al. (2015)* focused on bacteria producing biosurfactants tolerant to high salt concentrations, with potential applications in biotechnology. Soil and sediment samples collected from Trindade Island yielded 14 bacterial strains, mostly within the genus *Bacillus* (including *Bacillus subtilis* and its subspecies *Bacillus subtilis subsp. subtilis* and *Bacillus subtilis subsp. spizizenii*). Some strains remained unidentified, suggesting the possibility of novel species. These bacteria produced effective biosurfactants even at salt concentrations as high as 175.0 g L$^{-1}$; strains TR13 and TR14 were particularly successful in reducing interfacial and surface tension. However, a genetic analysis by

*Dunlap, Bowman & Zeigler (2020)* proposed that the four *Bacillus subtilis* subspecies (*subsp. subtilis*, *subsp. spizizenii*, *subsp. inaquosorum*, and *subsp. estercoris*) may warrant reclassification as separate species due to differences in their production of bioactive secondary metabolites. These findings underscore the complexity and wide-ranging biotechnological potential of the genus *Bacillus*.

Overall, the ability of these microorganisms to reduce interfacial tension and withstand challenging conditions suggests considerable promise for bioremediation and enhanced oil recovery in industrial and contaminated environments.

## Comparison between microbial communities of Trindade Island and the Galápagos Islands

In the study by *Zhao (2019)*, 48 environmental samples collected in 2015 and 2016 at 23 stations throughout the Galápagos archipelago were analyzed. Filters of 3 and 0.22 μm pore size were used to collect bacterial cells, and environmental parameters included particulate organic carbon (POC), dissolved organic carbon (DOC), chlorophyll *a*, phosphate, nitrate, silicate, and cell abundance measured by flow cytometry. Samples were sequenced using the Illumina HiSeq platform, allowing both taxonomic composition and ecological functions to be assessed based on unassembled reads.

Taxonomic analyses revealed communities dominated by bacteria, cyanobacteria, and archaea. Among the primary bacterial groups, *Candidatus pelagibacter* (SAR11) was the most abundant taxon. This group plays a key role in the carbon cycle, recycling DOC in oligotrophic environments and thereby contributing to the sustainability of marine ecosystems. Other important genera included *Formosa*, *Polaribacter*, and *Tenacibaculum* (all within Flavobacteria), which were linked to upwelling events. These microorganisms are involved in organic matter degradation and nutrient cycling, particularly in areas of high primary productivity.

El Niño 2015 and La Niña 2016 climatic events exerted a substantial influence on microbial community composition (*Zhao, 2019*). During the La Niña event, stronger upwelling brought nutrient-rich deep water to the surface, leading to increases in taxa suited to eutrophic environments, such as Rhodobacterales HTCC2255 and Gammaproteobacteria HTCC2207, alongside archaea like *Candidatus nitrosopelagicus*. This archaeon, dominant in some high-productivity areas, plays a central role in the nitrogen cycle through ammonia oxidation, especially in deeper waters. By contrast, the El Niño 2015 event weakened upwelling, resulting in more homogeneous microbial communities and relatively stable DOC and nutrient levels.

Cyanobacteria were pivotal to primary productivity, notably *Synechococcus* and *Prochlorococcus*. *Synechococcus* was associated with high-productivity sites (*e.g.*, Darwin Bay, where chlorophyll *a* increased), whereas *Prochlorococcus* was predominant in more oligotrophic eastern regions. Both genera directly contribute to carbon fixation and support the marine food chain (*Zhao, 2019*).

Whereas the microbial communities of Trindade Island were dominated by Bacteroidota, Actinobacteriota, and Pseudomonadota, Galápagos communities were marked by *Candidatus pelagibacter* (SAR11), *Synechococcus*, and *Prochlorococcus*. In the

Galápagos, archaea such as *Candidatus nitrosopelagicus* significantly influence the nitrogen cycle, a feature less pronounced in Trindade Island communities. Additionally, climatic events like El Niño and La Niña notably shaped Galápagos microbial assemblages through direct impacts on nutrient availability *via* upwelling, whereas on Trindade Island, microbial patterns appear to be primarily driven by local processes such as benthic–pelagic coupling and exudates released by turf algae.

These disparities demonstrate how microbial communities are molded by unique environmental conditions in each region, underscoring their essential role in the ecological resilience and sustainability of marine ecosystems.

## Comparison between microbial and fungal communities of Trindade Island and the Cuatro Ciénegas Basin

*Souza et al. (2018)* analyzed microbial communities from the Churince system in the Cuatro Ciénegas Basin (CCB), Mexico—a closed hydrological system with extremely high microbial diversity—by sampling water, sediment, and soil. Environmental DNA underwent 16S rRNA gene sequencing *via* 454 pyrosequencing, revealing 60 bacterial phyla and three archaeal phyla.

Aquatic environments in this system were dominated by Pseudomonadota, Actinobacteriota, and Bacteroidota, while sediments and soils showed higher overall diversity. The latter included Firmicutes and Cyanobacteria, which play key roles in nitrogen fixation and nutrient cycling. In sediments, cyanobacteria form microbial mats, whereas in soils, they participate in microbial crusts along with Acidobacteriota and Nitrospirota, reflecting localized adaptations.

Phylogenetic reconstructions of the genus *Bacillus* indicated endemic lineages in sediment and marine environments dating from the Ediacaran and Jurassic (*Souza et al., 2018*). These lineages perform crucial nutrient-recycling functions, exhibiting metabolic cooperation and local antagonism mechanisms that ensure survival under harsh conditions.

The CCB is often described as a microbiological "lost world," not only because of its high biodiversity but also its ancient microbial lineages, limited horizontal gene transfer, and adaptations to extreme nutrient imbalance. These factors create ecological barriers that prevent even similarly adapted taxa from successfully colonizing the basin.

Both the CCB and Trindade Island harbor high microbial diversity, supported by eco-evolutionary adaptations to extreme environments. Yet, while the CCB's microbial diversity is shaped by geologically stable, nutrient-poor, and often saline conditions—with notable phyla including Cyanobacteria, Pseudomonadota, Actinobacteriota, and Firmicutes—Trindade Island's microbial communities thrive in organic-rich, acidic soils, featuring phyla such as Actinobacteriota, Pseudomonadota, Acidobacteriota, Bacteroidota, and Verrucomicrobiota, alongside diverse fungal groups (notably Ascomycota and *Mortierella humilis*).

In the CCB, adaptations center on coping with nutrient scarcity, with cyanobacteria facilitating nitrogen fixation and forming microbial crusts. On Trindade Island, microbial

processes include decomposing organic matter—particularly plant fibers from giant ferns—highlighting the importance of Actinobacteriota in rich, acidic soils. Additionally, Pseudomonadota and Verrucomicrobiota play pivotal roles in nitrogen fixation and carbon cycling.

A striking contrast is the *Bacillus* genus in the CCB, which includes lineages dating to the Ediacaran and Jurassic, adapted to extremely oligotrophic habitats, compared to the predominance of *Rhodococcus* and *Nocardia* (both Actinobacteriota) on Trindade Island, noted for metabolizing complex hydrocarbons in an environment lacking historical anthropogenic contamination. The phylum Acidobacteriota, prevalent in Fazendinha soils, is vital for mineralization in nutrient-poor habitats, whereas Bacteroidota, found in reefs and soils, assists in organic-matter degradation.

*Velez et al. (2016)* collected water, sediment, and wood panel samples from three freshwater systems (Churince, Pozas Rojas, and Becerra) in the CCB, sequencing internal transcribed spacer (ITS) regions to investigate active fungal communities. The resulting fungal profiles showed moderate diversity, with lignocellulolytic and saprophytic anamorphic fungi dominating. Genera such as *Alternaria*, *Cladosporium*, *Stachybotrys*, *Trichoderma*, *Phoma*, and *Acremonium* played crucial roles in decomposing plant detritus and cycling nutrients under oligotrophic conditions. Species adapted to extreme environments (*e.g.*, *Acremonium persicinum* and *Emericellopsis pallida*) thrived due to elevated salinity in the CCB.

The genus *Stachybotrys*—often associated with perennial grasses like *Phragmites australis*—exhibited strong cellulolytic capacities and acted as a primary saprobe in plant litter. *Phoma*, widespread in the environment, has both saprophytic and phytopathogenic phases, while *Cladosporium* (also broadly distributed) includes saprophytic and phytopathogenic taxa capable of tolerating various stressors. Local lignocellulose colonization was largely governed by fungi such as *Volutella*, *Trichoderma*, and *Phoma*, which exhibit strategies to compete intensely for available substrates (*Velez et al., 2016*).

By contrast, Trindade Island's soil fungal communities are dominated by the phyla Ascomycota, Basidiomycota, and Mucoromycota. These fungi are adapted for carbon recycling under acidic, cooler conditions, as exemplified by *Antarctomyces psychrotrophicus*. *Mortierella humilis*—particularly prevalent in Fazendinha (PF7) soils—demonstrates adaptive capacity to organic-rich soils by decomposing complex compounds and forming mycorrhizal associations.

Despite differences in their respective habitats, the fungal communities of both the CCB and Trindade Island display specific adaptations to local environmental constraints. Whereas the CCB's fungi are specialized for oligotrophic, often saline conditions, the fungi on Trindade Island thrive in organic-rich, acidic soils. In each setting, fungi are vital for nutrient cycling, sustaining key ecological processes.

The differences and similarities between the microbial and fungal communities of Trindade Island and the CCB illustrate how these organisms have evolved to occupy distinct but equally challenging ecological niches. In both ecosystems, they play a fundamental role in maintaining ecological resilience by supporting critical biogeochemical processes. Their diverse adaptations likewise provide insights into

ancestral evolutionary pathways and interspecies interactions, pointing to the biotechnological potential of these communities, which holds considerable promise for biodiversity conservation and novel environmental applications.

### Limitations and future perspectives

The reviewed studies highlight the richness and complexity of microbial and fungal communities on Trindade Island, illustrating how environmental factors affect the health of marine and terrestrial ecosystems. These findings have important implications for understanding natural mechanisms of habitat management and restoration on Trindade Island and in comparable regions.

Despite this progress, there are intrinsic limitations in molecular identification techniques, particularly for bacterial, archaeal, and fungal taxa. For instance, numerous taxa detected in soil samples collected at higher elevations such as Pico do Desejado and Fazendinha could not be assigned to specific genera or species. This limitation likely arises from the high number of undescribed microbial species and gaps in existing taxonomic databases (*Konstantinidis, Rosselló-Móra & Amann, 2017*; *Lu et al., 2017*; *Marcelino et al., 2020*; *Seol et al., 2019*). Furthermore, there were also potential biases due to distinct sequencing platforms, sequencing depths and coverage, and mainly, the lack of temporal (longitudinal) studies, which could indeed clarify aspects concerning how seasonality (summer × winter, or rainy × dry seasons) affects the microbial diversity in the different substrates and environments previously studied in the Trindade Island.

Another critical constraint was the lack of access to raw sequencing data (*e.g.*, FASTQ files), which were unavailable in any of the reviewed articles. As a result, the analyses for this review relied solely on results reported in the main text and other supplemental materials. This was particularly problematic when examining studies of microcosm-based experiments, since many did not present detailed abundance tables at the genus level despite discussing genus-level findings. Such omissions limited the scope of inter-study comparisons and restricted our ability to profile the described microorganisms more comprehensively.

While DNA-based research on environmental samples from Trindade Island remains relatively limited, there is clear growth in what might be termed the "molecular era." However, achieving a fuller understanding of the island's diverse habitats, microbial communities, and their ecological functions requires expanded, dedicated efforts. This is especially vital given the uniqueness of Trindade Island, about which much remains to be discovered.

## CONCLUSION

This systematic review offers a detailed overview of the principal bacterial, archaeal, and fungal communities identified in soil, water, and coral samples from Trindade Island. These samples were analyzed using multiple methodologies, including sequencing of uncultivated samples, microcosm-enhanced samples, and culture-based approaches. Because raw sequencing data were unavailable, all data presented here were

derived exclusively from the selected articles' main texts and other supplemental materials. Nonetheless, data were standardized before analyses, and the integrated results emphasize pronounced differences in microbial and fungal diversity across ecological niches shaped by distinct environmental factors.

Marine microbial communities at Trindade Island exhibit high diversity, influenced by primary productivity and nutrient flux, particularly on the Trindade Shelf. In contrast, reefs dominated by turf algae contain high dissolved organic carbon (DOC), favoring bacteria adapted to these conditions while exacerbating hypoxia and coral stress. The soils of Pico do Desejado and Fazendinha host microbial communities specialized for acidic conditions and organic-rich environments, with key roles in carbon and nitrogen cycling. Fungal species such as *Mortierella* also contribute to complex compound decomposition, helping maintain soil fertility. Microcosm-enriched communities showed remarkable resilience and degradation capabilities for recalcitrant hydrocarbons, indicating robust metabolic potential under various contamination scenarios.

Compared to oligotrophic microbial assemblages in the Galápagos, which emphasize carbon recycling and nitrogen fixation, Trindade Island's communities appear shaped by local nutrient inputs that support a broader range of species. In the Cuatro Ciénegas Basin (CCB), microbial and fungal diversity reflects extreme oligotrophic and saline conditions, which sharply contrasts with the acidic, organic-rich soils of Trindade Island, despite both habitats' hosting unique microbial taxa.

Overall, this review underscores the importance of investigating Trindade Island's microbial and fungal assemblages from diverse angles, given their relevance for biogeochemical cycles and ecological stability. Moreover, these communities hold promise for biotechnological applications such as bioremediation and ecosystem management. Future work should intensify efforts to apply state-of-the-art sequencing technologies and freely share raw data for deeper comparative analyses. Our research group is currently conducting an integrative study of taxonomic and functional diversity using shotgun metagenomics, paired with comprehensive physicochemical characterizations of different vegetation zones on this remarkable and remote South Atlantic island.

## ACKNOWLEDGEMENTS

We would like to thank the Graduate Programmes of Bioinformatics and Microbiology of the Universidade Federal de Minas Gerais (UFMG), ProRectory of Research of Federal University of Minas Gerais, Centro Federal de Educação Tecnológica de Minas Gerais (CEFETMG), Universidade Federal de Santa Catarina (UFSC), Universidade Estadual de Feira de Santana (UEFS), Universidade Federal de Viçosa (UFV), and Universidad de San Francisco Xavier de Chuquisaca (USFX).

### Funding

This work was supported by CNPq (Conselho Nacional de Desenvolvimento Científico e Tecnológico): Funga da Ilha da Trindade: revelando a diversidade escondida

(443316/2019-8) and Do meio do oceano para a nuvem digital: Bio e Ecoinformática da Funga da Ilha da Trindade, Brasil (440020/2020-4). The funders had no role in study design, data collection and analysis, decision to publish, or preparation of the manuscript.

## Grant Disclosures

The following grant information was disclosed by the authors:
Conselho Nacional de Desenvolvimento Científico e Tecnológico (CNPq): 443316/2019-8 and 440020/2020-4.

## Competing Interests

The authors declare that they have no competing interests.

## Author Contributions

- Glen Jasper Yupanqui García conceived and designed the experiments, performed the experiments, analyzed the data, prepared figures and/or tables, authored or reviewed drafts of the article, and approved the final draft.
- Fernanda Badotti conceived and designed the experiments, performed the experiments, analyzed the data, authored or reviewed drafts of the article, and approved the final draft.
- Alice Ferreira-Silva conceived and designed the experiments, performed the experiments, analyzed the data, authored or reviewed drafts of the article, and approved the final draft.
- Joyce da Cruz Ferraz Dutra conceived and designed the experiments, performed the experiments, analyzed the data, prepared figures and/or tables, authored or reviewed drafts of the article, and approved the final draft.
- Kelmer Martins-Cunha conceived and designed the experiments, performed the experiments, analyzed the data, prepared figures and/or tables, authored or reviewed drafts of the article, and approved the final draft.
- Rosimeire Floripes Gomes conceived and designed the experiments, performed the experiments, analyzed the data, authored or reviewed drafts of the article, and approved the final draft.
- Diogo Costa-Rezende conceived and designed the experiments, performed the experiments, analyzed the data, authored or reviewed drafts of the article, and approved the final draft.
- Thairine Mendes-Pereira conceived and designed the experiments, performed the experiments, analyzed the data, authored or reviewed drafts of the article, and approved the final draft.
- Carmen Delgado Barrera conceived and designed the experiments, performed the experiments, analyzed the data, authored or reviewed drafts of the article, and approved the final draft.
- Elisandro Ricardo Drechsler-Santos conceived and designed the experiments, performed the experiments, analyzed the data, authored or reviewed drafts of the article, and approved the final draft.

- Aristóteles Góes-Neto conceived and designed the experiments, performed the experiments, analyzed the data, authored or reviewed drafts of the article, and approved the final draft.

## Data Availability

This is a systematic review/meta-analysis.

## Supplemental Information

Supplemental information for this article can be found online at http://dx.doi.org/10.7717/peerj.19305#supplemental-information.

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
