# Peer review of "Microbial diversity of the remote Trindade Island, Brazil: a systematic review"

_PeerJ, doi:10.7717/peerj.19305_

## Round 0.1 · original submission · Major Revisions

Please provide a thorough revised manuscript and a point-by-point rebuttal letter. Detailed comments will help improve your document.

·

Basic reporting

The manuscript presents a systematic meta-review of microbial diversity on Trindade Island, focusing on the ecological roles and biotechnological potential of Archaea, Bacteria, and Fungi. It synthesizes a wide range of existing literature, emphasizing the island’s unique microbial ecosystems and their potential applications in environmental conservation and bioremediation.

While the manuscript covers a substantial amount of data, it would benefit from significant revisions to improve the structure and clarity of the findings. Currently, the results are detailed but tend to be overly descriptive rather than analyzed and synthesized. This approach makes it difficult to identify key insights and understand their relevance. A more concise presentation, focusing on the most important outcomes, would make the narrative clearer and more impactful. Streamlining the content and concentrating on the key results would enhance both the readability and the overall impact of the manuscript.

1. The title gives the impression that the study is presenting new data, which is misleading. Since this is a meta-review, it should clearly reflect that the paper synthesizes existing research rather than reporting on new experimental work. Additionally, while the study touches on biotechnological potential, this aspect seems secondary to microbial diversity. The title should accurately represent the primary focus of the manuscript to avoid misleading readers about the study’s emphasis.

2. In the abstract and Introduction, the hypothesis or study question is not What specific research questions were you aiming to answer, and what were the primary hypotheses driving this study?

3. The introduction outlines the objective to "synthesize and interpret" microbial diversity on Trindade Island but does not present a formal hypothesis or guiding research question. A clearly defined hypothesis would provide stronger direction and focus for the manuscript. Without this, the study lacks the specificity required to tie the various elements together into a cohesive argument. Additionally, the introduction contains general and basic information (see attached revised PDF) that does not align closely with the study's specific focus. For example, sections like lines 76-79 and 85-96 could be removed without losing any essential context or argument. Instead of these generic details, the introduction should be refocused to address the knowledge gaps and the necessity of synthesizing available studies on Trindade Island's microbial communities. Incorporating more relevant literature or background information that highlights the importance of these questions will better justify the need for this meta-review and underscore the potential insights that reinterpreting existing data could reveal..

3. The Results and Discussion sections need clearer separation and organization. Currently, these sections blur together and are highly descriptive, which makes it difficult for readers to navigate and understand the most important findings. The manuscript would benefit from a structured synthesis of the data, breaking down findings by ecological niche (e.g., soil, water, coral) and summarizing key trends and insights across microbial taxa. This would provide a more focused and digestible narrative. Additionally, reducing redundancies and streamlining the descriptions would help avoid overwhelming the reader with exhaustive taxonomic lists.

4. The manuscript would be greatly improved by comparing Trindade Island’s microbial communities to those from other polluted or isolated environments, such as oceanic islands facing similar environmental challenges (e.g., oil pollution, limited biodiversity, isolation). Additionally, comparisons to unperturbed, salt-rich regions like Atacama or Cuatro Ciénegas would provide valuable context for interpreting the findings. Such comparisons are crucial for assessing whether the microbial patterns observed on Trindade Island are unique or consistent with other ecosystems.

5. The conclusion highlights some interesting biological and ecological findings but does not clearly reflect the meta-review nature of the study. The phrasing in some places gives the impression that the manuscript is reporting new data rather than synthesizing existing findings. For example, the statement "Our results indicate a cryptic microbial diversity" should be rephrased to something more accurate, like "Our review indicates" or "The analysis of reviewed studies suggests" to avoid confusion about the origin of the findings.

6. Figure 2 does not add substantial value to the manuscript. It is referenced only briefly, indicating that it does not play a critical role in the analysis. If the intention of the figure is to highlight the profile of the studies included in the meta-review, a Venn diagram or pie chart would more effectively illustrate the overlap and unique aspects of different sample types, methodologies, or study focuses. Additionally, the use of bright colors and the island-map shape makes the figure harder to interpret and more suitable for a presentation than a peer-reviewed paper.

While the manuscript presents valuable data, its cohesion is compromised by structural issues, excessive detail, and a lack of integration across themes. By refining the structure, synthesizing the data more effectively, and providing comparative context, the manuscript could become more unified and impactful. With these revisions, the manuscript has the potential to be much clearer and make a more significant contribution to the field.

Experimental design

1. Section "Idiosyncrasies of the Selected Studies" is not an ideal representation of the content. The content is mostly about the methodological approaches (e.g., culture-dependent vs. culture-independent techniques), technological limitations, and the challenges of taxonomic classification. The title should better reflect that the section is about summarizing methodological challenges, data limitations, and opportunities for future research.

2. Marker-based taxonomic identification methods, such as 16S rRNA sequencing, offer genus-level resolution but do not provide detailed insights into the metabolic pathways, enzymes, or functional capabilities of the identified organisms. Therefore, making definitive claims about their biotechnological potential (e.g., pollutant degradation, bioremediation) based solely on these identifications is speculative without the support of further functional assays or experiments.

3. Furthermore, ecovariants of the same species can exhibit drastically different metabolic capacities depending on factors like environmental conditions, gene regulation, or horizontal gene transfer. For instance, while one strain of Rhodococcus rhodochrous may be highly efficient at degrading hydrocarbons, another strain of the same species may lack the necessary metabolic pathways entirely. This variability emphasizes the risks of overgeneralizing functional capabilities based on partial genetic data, especially when metadata on physicochemical conditions at the time of sampling is unavailable. Without this context, it is challenging to accurately predict the functional roles of the microbial communities.

4. Grouping by broad environmental categories (e.g., soil, water, coral) is common in microbial ecology studies, and the patterns observed in PCoA are likely to be similar across other ecosystems (such as Atacama in Chile, Cuatro Cienegas in Mexico or Sharkbay in Australia). This reduces the novelty and uniqueness of the findings for Trindade Island, making the study appear more like a general ecological survey than a deep exploration of the island's distinct microbial feature. To provide more meaningful and novel information, the authors should consider finer-scale analyses that focus on a gradient of pollution or salt-concentration.

Validity of the findings

- The manuscript highlights areas where microbial diversity is understudied, especially in terms of functional diversity and species-level identification. These gaps are valid and important for future research. The study provides reliable insights into taxonomic diversity, especially at higher taxonomic levels (phylum, genus). The identification of various Archaea, Bacteria, and Fungi is sound based on the available data, even if more detailed insights are limited by the use of marker genes.

- However, taxonomic findings are generally valid but limited to descriptive insights.
- How do authors address the limitations in standardization of data from different studies (e.g., different methods for analyzing the community (culture dependent and independent data), taxonomic databases, diversity indexes, methodologies and biases in taxonomic bining of data)?

- The reliance on culture-dependent studies for some of the data further limits the findings. Culture-dependent methods can exclude a significant portion of microbial diversity, particularly in extreme or unique environments like Trindade Island. This introduces bias into the results, reducing the validity of claims about microbial community structure.

- The manuscript does not seem to provide sufficient follow-up with functional metagenomic data or experimental validation of the biotechnological potential, which would be critical to substantiate these claims. Simply identifying a species or strain from marker genes does not guarantee the presence of functional pathways required for the activities being proposed.

- The findings are limited by the absence of whole-genome sequencing or functional metagenomics. While the study suggests bioremediation potential, the actual metabolic capabilities of the microbes in question are not directly demonstrated or found in the review, which undermines the claims of biotechnological relevance.

Additional comments

Without a thorough understanding of the specific functional gene content of these isolates—through techniques such as shotgun metagenomics, transcriptomics, or proteomics—the claims regarding pollutant degradation and other industrial applications are premature. While these microbial communities may have been enriched by selective pressures from pollutants, making the description of enriched populations relevant, the functional capabilities of these microbes remain speculative.

The authors should explicitly address the limitations of using partial marker gene sequencing to predict biotechnological potential. Rather than presenting these speculative claims as confirmed findings, they should frame them as suggestions for further exploration. Additionally, speculative claims should not be highlighted prominently in the title and abstract, as this could mislead readers regarding the certainty of the findings.

Reviewer 2 ·

Basic reporting

This manuscript presents a re-analysis that synthesizes data on the microbial communities of Trindade Island, aiming to provide valuable insights for biodiversity conservation and biotechnological applications. The authors have analyzed filtered data from eight previous research papers, comparing their taxonomical classifications and alpha and beta diversity analyses.

The study design and findings are clearly presented. However, the order of the sections may confuse readers. Specifically, the "Study Selection" and "Profile of Selected Articles" sections should follow the "Data Collection" section for a more logical flow. The figures are clear and concise, though I recommend increasing the font size of Figure 6 for better readability.

Experimental design

Given the experimental variations between the datasets from the different studies, re-analyzing the sequences under uniform parameters would yield more meaningful results. This is particularly important due to differences in sequencing library preparation, bioinformatics pipelines used for taxonomic classification, and the databases employed. At a minimum, studies employing amplicon-based approaches should be re-analyzed using consistent parameters, despite variations in hypervariable regions and sequencing platforms.

Additionally, contrasting microbial abundances between groups using a more formal statistical analysis, such as LefSe or similar, could help identify overabundant taxa and provide a stronger biological context. Moreover, leveraging the reanalyzed data to infer metabolic capabilities using tools like PICRUSt or similar would provide deeper insights into the functional potential of these communities.

For full transparency and to facilitate future analyses, it would be beneficial to include the code used to generate the results. Lastly, I suggest including the specific details of each dataset (e.g., hypervariable regions used) in Table 2 to enhance clarity and context.

Validity of the findings

The data analysis and comparisons are well-executed, with a clear objective. However, the conclusions drawn in this manuscript do not extend significantly beyond those presented in the original research papers analyzed. The authors should ensure the conclusions reflect the limitations inherent in the re-analysis and offer more nuanced insights based on this synthesis.

Addressing these recommendations will improve the manuscript and contribute to its successful publication.

Reviewer 3 ·

Basic reporting

The authors do an important and thorough review on the microbiological findings on samples from Trindade Island DNA sequencing and molecular methods.
The methods for literature screening are sound and well documented and the studies selected were well explored.
There is a challenge on making compatible data from different studies, conducted with different criteria and reported in different manners. And the authors managed to tackle this challenge gracefully. Ideally, reviews of this nature would be conducted acquiring the raw sequencing data and re-analyzing it to construct new tables and make us of the latest reference taxonomic databases. However, in this case, considering the little number of studies devoted to Trindade Island and the variability of the openness of the data, I believe the authors did a great job in combining and reporting them.
I found misleading the way the authors use the term culture-dependent approaches. For instance:
At line 150 and 151, when establishing a system for grouping the samples, the authors say that 1 sample named Crude Oil was identified as Soil_DCul and 12 samples were identified as Water_DCul. However, those samples were assessed with metabarcoding analyses and although an experiment was used to assess the microbial communities, they were not culture-dependent, in the sense that there was no need to grow those microbes for assessing the community.
Moreover, at line 414, it says that 6 studies were used, but at line 196 it says 8 studies were used.

I like the idea of making a Venn diagram to show commonalities between observations, but I see the limitations of using such graphical resource when having many groups. My suggestions would be to use UpSet plots (https://upset.app/) instead.

One last suggestion would be to add a table describing what were the methods used in those studies. Bioinformatics pipelines, diversity metrics used, normalization/scaling methods (rarefaction/RLE/TMM/etc), statistical tests applied to compare samples and not only the list of ribosomal regions, but a list of primers used by the metabarcoding approaches.

In summary the study is very relevant and the data was well explored.

Experimental design

No comment.

Validity of the findings

When reporting the Soil_DCul. The sequencing for that experiment was done for control samples and oil contaminated samples. But the authors report on a single sample at line 151. Was that the mean value from both treatments? This question expands for all the other studies, how were the numbers acquired?
I would suggest separating the Control and Oil into two samples, as this would reflect "natural" soils from Trindade and the response of the microbial community after oil contamination. This would also change the results of "Distribution of Fungi in the Samples" as the control of Soil_DCul has Ascomycota as predominant (Line 282) in the control and Mucoromycota as predominant in Oil contaminated samples.

Additional comments

I noticed that the Sample Soil_DCul doesn't appear in the leaflet visualization map. The precise location can be acquired in contact with the authors of Morais et al., 2016 (20°29'45.5"S 29°19'44.2"W).

---

## Round 0.2 · Minor Revisions

Please address all reviewer feedback. Additionally, ensure that grammar and style are corrected. Neglecting this may necessitate another round of revisions. If you utilized a professional editing service, kindly provide proof of that.

**Language Note:** The Academic Editor has identified that the English language must be improved. PeerJ can provide language editing services - please contact us at [email protected] for pricing (be sure to provide your manuscript number and title). Alternatively, you should make your own arrangements to improve the language quality and provide details in your response letter. – PeerJ Staff

·

Basic reporting

The manuscript is written in professional and intelligible English, maintaining a formal tone appropriate for academic work. While the language is generally clear, there are instances where complex sentence structures and technical jargon could benefit from simplification or additional explanation to ensure accessibility for a broader audience. Overall, the text effectively communicates the study’s objectives, methodology, and findings. The manuscript is self-contained, providing sufficient background, detailed methods, and relevant results that address the stated hypotheses. The systematic approach, guided by PRISMA standards, ensures that the research is comprehensive and grounded in methodological rigor. The results align well with the research focus on microbial diversity and ecological roles, but the discussion of biotechnological potential is limited, which may leave certain hypotheses underexplored. Visual elements such as figures and tables are well-designed and contribute meaningfully to the presentation of data. With minor refinements to simplify language of legends/captions and better align results with hypotheses, the manuscript will be further enhanced in readability and coherence.

Experimental design

The manuscript aligns well with the aims and scope of PeerJ, focusing on the ecological and microbial diversity of a unique and understudied ecosystem, which contributes to advancing knowledge in environmental microbiology and conservation. The research question is well-defined, relevant, and meaningful, addressing microbial diversity and ecological roles on Trindade Island while identifying knowledge gaps, such as limited studies on archaea and fungi in this environment. The methods are described in sufficient detail, including the use of PRISMA guidelines, comprehensive taxonomic analyses, and statistical approaches, allowing for potential replication of the study.

Validity of the findings

The manuscript addresses a clear, relevant, and meaningful research question that centers on microbial diversity and the ecological roles of microorganisms in Trindade Island's unique volcanic environment. This focus is especially significant due to the island's ecological importance, isolation, and the broader interest in understanding microbial contributions to ecosystem resilience and potential biotechnological applications. The study successfully fills a critical knowledge gap by synthesizing fragmented information from prior research, providing a thorough overview of microbial diversity across diverse habitats. It also identifies key taxa involved in ecological processes and contextualizes these findings within global biodiversity patterns by comparing Trindade Island with other ecosystems, such as the Galápagos Islands and Cuatro Ciénegas Basin.

However, the connection between microbial diversity and practical applications, particularly biotechnological ones, could be further developed. While interesting and necessary to mention, this aspect should be presented as a secondary observation rather than a central focus of the paper, especially given that the study predominantly relies on 16S rRNA analysis, which limits functional characterization and biotechnological insights.

Additionally, the manuscript would benefit from a more explicit discussion of its limitations. These include potential sampling biases due to the focus on specific regions of the island, technological constraints from older sequencing methods, the lack of temporal and seasonal variation in the data, and limited taxonomic resolution with a substantial proportion of unclassified taxa. Briefly addressing these issues would enhance the robustness and generalizability of the findings while providing clear directions for future investigation.

Overall, the manuscript effectively tackles an understudied topic and establishes a valuable foundation for future research on microbial diversity in extreme and isolated environments. Its relevance extends to both ecological studies and applied sciences.

Additional comments

The current title, "Microbial Diversity and Biotechnological Potential of the Remote Trindade Island, Brazil: A Systematic Review," positions "biotechnological potential" as a central focus. However, the manuscript dedicates only a single paragraph to this topic and lacks a detailed or focused analysis of biotechnological applications. The majority of the content emphasizes microbial diversity, taxonomic composition, and ecological roles, with minimal discussion of biotechnological aspects. This imbalance may give readers an inaccurate impression of the manuscript’s scope and depth in addressing biotechnological potential.

Furthermore, the figures and tables in the manuscript are dedicated to analyzing and supporting the microbial diversity of Trindade Island. None of the visual or data elements are focused on illustrating biotechnological applications. While this is not a limitation of the manuscript itself, it further supports the recommendation to revise the title to better reflect the primary focus of the study. A suggested revision is: "Microbial Diversity of the Remote Trindade Island, Brazil: A Systematic Review." The final decision on the title, however, rests with the editor.

Reviewer 2 ·

Basic reporting

The paper presents clear and unambiguous results that are self-contained, directly relevant to the stated objectives, and easy to follow. The necessary figures effectively support the findings. However, I have the following suggestions to improve the presentation of the figures:
* In Figure 4, the font size is difficult to read. I suggest increasing it for better readability.
* For the alpha diversity analysis, it would be helpful to display significant differences between groups directly within the boxplots to emphasize their distinctions.
* Additionally, including a metric to compare the richness of each group (such as Observed Genus or the lowest taxonomic level possible) would add valuable context. A dedicated plot for this metric would be beneficial.

Experimental design

The research question is well-defined with sufficient detail, despite the missing information for the meta-analysis. It is clear that the authors lack access to the available datasets for reanalysis, but the experimental design is robust, allowing the results to be well-supported and contained.

Validity of the findings

The conclusions are well-stated, appropriately supported by the results, and provide a clear perspective for future research.

Additional comments

I am pleased to see that the authors have diligently addressed all the suggestions and corrections made by the reviewers in this revised version of the manuscript. The improvements have significantly enhanced the overall quality of the article, and I commend the authors for their efforts. I have no further revisions to suggest.

Reviewer 3 ·

Basic reporting

.

Experimental design

.

Validity of the findings

.

Additional comments

* Line 78 - "reducing the abundance of fungi and changing the composition of bacterial phyla (Morais et al., 2016)"
Abundance of fungi wasn't evaluated at Morais et al., 2016, it was meant perhaps "reducing the number of fungal taxa".

The geolocation in the map for sample "Sample soil", taken for the treatment crude oil, in the leaflet online map is incorrect. The actual sampling location is the North West of the Island (20°29'45.5"S 29°19'44.2"W) as in the attached image.

Annotated reviews are not available for download in order to protect the identity of reviewers who chose to remain anonymous.

---

## Round 0.3 · accepted · Accept

Congratulations! Your contribution to PeerJ has now been accepted!